# `CoDe`: **Blockwise Control for Denoising Diffusion Models**

**Anuj Singh**[1 2]        **Sayak Mukherjee**[1]        **Ahmad Beirami**[3]        **Hadi Jamali-Rad**[1 2]

[1]**Delft University of Technology, The Netherlands**
[2]**Shell Global Solutions International B.V., Amsterdam, The Netherlands**
[3]**Massachusetts Institute of Technology, Cambridge MA, USA**

`{a.r.singh, h.jamalirad}@tudelft.nl`

**Reviewed on OpenReview:** *https://openreview.net/forum?id=DqPCWMiMUO*

## Abstract

Aligning diffusion models to downstream tasks often requires finetuning new models or gradient-based guidance at inference time to enable sampling from the reward-tilted posterior. In this work, we explore a simple inference-time gradient-free guidance approach, called controlled denoising (`CoDe`), that circumvents the need for differentiable guidance functions and model finetuning. `CoDe` is a blockwise sampling method applied during intermediate denoising steps, allowing for alignment with downstream rewards. Our experiments demonstrate that, despite its simplicity, `CoDe` offers a favorable trade-off between reward alignment, prompt instruction following, and inference cost, achieving a competitive performance against the state-of-the-art baselines. Our code is available at: *https://github.com/anujinho/code*

## 1 Introduction

Diffusion models have emerged as a powerful tool for generating high-fidelity realistic images, videos, natural language content and even molecular data (Ho et al., 2020; Song et al., 2020; Bar-Tal et al., 2024; Wu et al., 2022). While diffusion models have proved to be effective at modeling complex and realistic data distributions, their successful application often hinges on following user-specific instructions in the form of images, text, bounding-boxes or other *rewards*. A common approach to the *alignment* of diffusion models to user preferences involves finetuning them on preference data, which

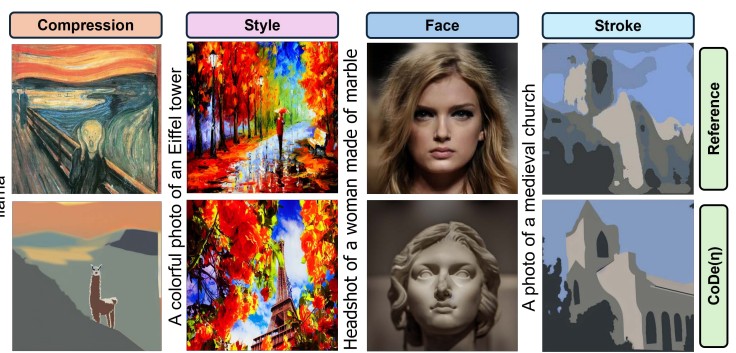

Figure 1: `CoDe` generates high quality compression (non-differentiable reward), style, face and stroke (differentiable rewards) guided images.

is typically done through reinforcement learning (RL), to generate samples with a higher reward while maintaining a low KL divergence from the base diffusion model (Fan et al., 2023; Uehara et al., 2024a).

Guidance-based approaches keep the base diffusion model frozen and control its output by aligning its generative process to a reward function at inference time. *Gradient-based guidance* methods utilize gradients of the reward at each denoising step to align the generated samples with the downstream task (Chung et al., 2023; Yu et al., 2023; Bansal et al., 2024b; He et al., 2024). In addition to requiring access to a *differentiable* reward signal, these approaches require memory-intensive gradient computations. On the other hand, *gradient-free guidance* methods such as Best-of-$N$ (Beirami et al., 2024) circumvent the need for differentiable rewards but can potentially be computationally intractable as they sometimes need a large number of samples, $N$, to satisfy the alignment goal.

In this paper, we consider a simple gradient-free guidance approach that aims at remedying the intractability of best-of-$N$. Drawing inspiration from blockwise controlled decoding in language models (Mudgal et al., 2024), we propose controlled denoising (`CoDe`), which exerts best-of-$N$ control over $N$ blocks of $B$ denoising steps rather than waiting for the fully denoised images. Our *key contributions* can be summarized as follows:

I. We propose `CoDe` — an inference-time blockwise guidance approach which samples from an optimal KL-regularized objective. We study the interplay between the sample size ($N$) and block-size ($B$) and demonstrate that `CoDe` is effective at improving the reward at the cost of the least amount of KL divergence from the base model.

II. We assess the performance of the aligned diffusion models structurally for two case studies (Gaussian Mixture Model (GMM), and image generation), in four scenarios under image generation: style, face, stroke, and compressibility guidance. Our extensive (qualitative and quantitative) experimental results demonstrate that `CoDe` achieves competitive performance against the state-of-the-art baselines, while offering a balanced trade-off between reward alignment, prompt instruction following, and inference cost.

## 2 Preliminaries

### 2.1 Diffusion Models

A diffusion model provides an efficient procedure to sample from a probability density $q(x)$ by learning to invert a forward diffusion process. The forward process is a Markov chain iteratively adding a small amount of random noise to a "clean" data point $x_0 \in \mathcal{X}$ over $T$ steps. The noisy sample at step $t$ is given by $x_t = \sqrt{\bar{\alpha}_t}x_0 + \sqrt{1 - \bar{\alpha}_t}\varepsilon_t$, where $\varepsilon_t \sim \mathcal{N}(0,1)$, $\alpha_t = 1 - \beta_t$, $\bar{\alpha}_t = \prod_{t=1}^{T}\alpha_t$, and $\{\beta_t\}_{t \in [T]}$ is a variance schedule (Ho et al., 2020; Nichol & Dhariwal, 2021). The forward process can then be expressed as:

$$q(x_{1:T}|x_0) = \prod_{t=1}^{T} q(x_t|x_{t-1}), \quad q(x_t|x_{t-1}) = \mathcal{N}(x_t; \sqrt{1 - \beta_t}x_{t-1}, \beta_t\mathbf{I}). \tag{1}$$

Now, to estimate $q(x)$, the diffusion model $p_\theta$ learns the conditional probabilities $q(x_{t-1}|x_t)$ to reverse the diffusion process starting from a fully noisy sample $x_T \sim \mathcal{N}(0,1)$ as:

$$p_\theta(x_0) = p(x_T) \prod_{t=1}^{T} p_\theta(x_{t-1}|x_t), \quad p_\theta(x_{t-1}|x_t) = \mathcal{N}(x_{t-1}; \mu_\theta(x_t, t), \beta_t\mathbf{I}), \tag{2}$$

where the variance is fixed at $\beta_t\mathbf{I}$, and only $\mu_\theta(x_t, t)$ is learned as:

$$\mu_\theta(x_t, t) = \frac{1}{\sqrt{\alpha_t}}\left(x_t - \frac{1 - \alpha_t}{\sqrt{1 - \bar{\alpha}_t}}\varepsilon_\theta(x_t, t)\right). \tag{3}$$

Here, $\varepsilon_\theta$ is a neural network which attempts to predict the noise added to $x_{t-1}$ in the forward process as:

$$\varepsilon_\theta(x_t, t) \approx \varepsilon_t = \frac{x_t - \sqrt{\bar{\alpha}_t}x_0}{\sqrt{1 - \bar{\alpha}_t}}. \tag{4}$$

Furthermore, using a conditioning signal $c$, diffusion models can be extended to sample from $p_\theta(x|c)$. The conditioning signal, $c$, can take diverse forms, from text prompts and categorical information to semantic maps (Zhang et al., 2023; Mo et al., 2023). Our work focuses on a text-conditioned model, Stable Diffusion (Rombach et al., 2021), which has been trained on a large corpus consisting of $M$ image-text pairs $\mathcal{D} = \{(x^i, c^i)\}_{i=1}^{M}$ using a reweighed version of the variational lower bound (Ho et al., 2020) as optimization loss function

$$\hat{\theta} = \arg\min_\theta \mathbb{E}_{t \sim [1,T], \, x_0, \varepsilon_t}\left[\|\varepsilon_t - \varepsilon_\theta(\sqrt{\bar{\alpha}_t}x_0 + \sqrt{1 - \bar{\alpha}_t}\varepsilon_t, \, c, \, t)\|^2\right]. \tag{5}$$

## 2.2 KL-Regularized Objective

Consider we have access to a reference diffusion model $p(\cdot)$, which we refer to as the *base* model. Note that here we drop $\theta$ (from $p_\theta$) for the ease of notation, also because base diffusion model parameters are kept intact throughout the inference-time guidance. Our goal is to obtain samples from the base model that optimize a downstream reward function $r(\cdot) : \mathcal{X} \to \mathbb{R}$, while ensuring that the sampled data points do not deviate significantly from $p$ to prevent degeneration in terms of image fidelity and diversity of the output samples (Ruiz et al., 2023). Thus, we aim to sample from a reward *aligned* diffusion model ($\pi$) that optimizes for a KL-regularized objective to satisfy both requirements. Let us start by defining some key concepts.

**Value function**. The expected reward when decoding continues from a partially decoded sample $x_t$:

$$V(x_t; p) = \mathbb{E}_{x_0 \sim p(x_0 | x_t)}[r(x_0)]. \tag{6}$$

**Advantage function**. We can define a one-step advantage of using another diffusion model $\pi$ for optimizing the downstream reward as:

$$A(x_t; \pi) := \mathbb{E}_{x_{t-1} \sim \pi(x_{t-1}|x_t)}\left[V(x_{t-1}; p)\right] - \mathbb{E}_{x_{t-1} \sim p(x_{t-1}|x_t)}\left[V(x_{t-1}; p)\right]. \tag{7}$$

It is important to note that the advantage of the base model (when $\pi = p$) is 0. Thus, we aim to choose an *aligned* model $\pi$ to achieve a positive advantage over the base model.

**Divergence**. We further denote the KL divergence ($\mathrm{KL}(.||.)$, also known as relative entropy) between the aligned model $\pi$ and the base model $p$ at each intermediate step $x_t$ as:

$$D(x_t; \pi) := KL\big(\pi(x_{t-1}|x_t) \,\|\, p(x_{t-1}|x_t)\big) = \int \pi(x_{t-1}|x_t) \log \frac{\pi(x_{t-1}|x_t)}{p(x_{t-1}|x_t)} \, dx_{t-1}. \tag{8}$$

**Objective**. Using Eq. (7) and Eq. (8), we can now formulate the KL-regularized objective as:

$$\pi_\lambda^* = \arg\max_\pi \left[\lambda A(x_t; \pi) - D(x_t; \pi)\right], \tag{9}$$

where $\lambda \in \mathbb{R}^{\geq 0}$ trades off reward for drift from the base diffusion model $p$.

**Theorem 2.1.** *The optimal model $\pi_\lambda^*$ for the objective formulated in Eq. (9) is given by:*

$$\pi_\lambda^*(x_{t-1}|x_t) \propto p(x_{t-1}|x_t) \, e^{\lambda V(x_{t-1}; p)}. \tag{10}$$

As we shall discuss in Section 3, our proposed approach builds on Theorem 2.1 to approximately sample from this reward aligned model using a Monte Carlo sampling strategy. An extension of the result in a conditional diffusion setting can be found in Appendix A. Notably, this is a step-wise variant of the more widely known similar objective (Korbak et al., 2022), which has been used in the context of language models (Beirami et al. (2024); Mudgal et al. (2024)), and in some learning-based methods (Prabhudesai et al., 2023; Fan et al., 2023; Wallace et al., 2023; Black et al., 2023; Gu et al., 2024; Lee et al., 2024) discussed in Section 7 for fine-tuning a diffusion model. However, contrary to the prior art, we use our objective directly for a guidance-based alignment, where as the end-to-end objective would be intractable. We also remark that this advantage is similar to controlled decoding (Mudgal et al., 2024) and how it enables efficient sampling from reward guided distributions in language models. In Appendix B, we demonstrate that sampling can be achieved using Langevin dynamics (Welling & Teh, 2011), resulting in a generalized form of classifier guidance (Dhariwal & Nichol, 2021). However, a key limitation of gradient-based approaches is the need for a differentiable reward function. To alleviate this, we explore a sampling-based method for model alignment allowing us to handle both differentiable and non-differentiable downstream rewards.

## 3 CoDe: Blockwise Controlled Denoising

Inspired by recent RL-based alignment strategies for LLMs through process rewards or value-guided decoding (Mudgal et al., 2024), we propose a sampling-based guidance method to align a conditional pretrained diffusion model, $p(\cdot|c)$, following the optimal solution, $\pi_\lambda^*$, described in Theorem 2.1. In the following, we

outline an approach to approximate the value function for intermediate noisy samples followed by introducing our sampling-based alignment strategy. Our proposed approach, coined as CoDe, is summarized in Algorithm 1.

**Approximation of the value function.** To compute the value function in Eq. (6) for an intermediate noisy sample $x_t$, it is necessary to compute the expectation over $x_0 \sim p(x_0|x_t)$. Note that for diffusion models such as DDPMs (Ho et al., 2020), the predicted clean sample $\hat{x}_0$ can be estimated given an intermediate sample $x_t$ using Tweedie's formula (Efron, 2011) as follows:

$$\hat{x}_0 = \mathbb{E}[x_0|x_t] = \frac{x_t - \sqrt{1 - \bar{\alpha}_t}\varepsilon_\theta(x_t, c, t)}{\sqrt{\bar{\alpha}_t}}. \tag{11}$$

By plugging Eq. (11) into Eq. (6), the value function can be approximated as:

$$V(x_t; p, c) = \mathbb{E}_{x_0 \sim p_\theta(x_0|x_t,c)}[r(x_0)] \approx r(\mathbb{E}[x_0|x_t]) = r(\hat{x}_0). \tag{12}$$

The benefit of such an approximation is that it circumvents the need for training a separate model to learn the value function, as is for instance adopted by DPS (Chung et al., 2023) and Universal Guidance (Bansal et al., 2024b). However, in certain scenarios, it is also possible to use a pre-trained detection or segmentation model to extract reward-aligned features and then use Tweedie's formula to obtain $\hat{x}_0$. According to the Tweedie's formula, the approximation of the conditional expectation, $\mathbb{E}_{x_0}[r(x_0)]$, is tight when the base diffusion model parameters $\theta$ perfectly optimize Eq. 5. For example, this approximation is expected to be more accurate towards the end of the denoising process (Ye et al., 2024).

Our objective is to achieve an improved alignment vs. divergence trade-off by sampling from the optimal solution presented in Theorem 2.1. Therefore, by taking advantage of the approximation in Eq. (12), we present a blockwise extension of Best-of-N (BoN) for diffusion models, termed as **Co**ntrolled **De**noising (CoDe) and outlined in Algorithm 1. Note that the timesteps during diffusion denoising are indexed from $T$ to 0 (in descending order), and not 0 to $T$ (line 3). CoDe integrates BoN sampling into the standard inference procedure of a pretrained diffusion model. Unlike BoN, instead of rolling out the full denoising $N$ times and selecting the best resulting sample, we opt for performing blockwise BoN. Specifically, for each block of $B$ steps, we unroll the diffusion model $N$ times independently (Algorithm 1, line 5). Then, based on the

---

**Algorithm 1: CoDe**

---

**Require:** $p, T, N, B, c$

1  Sample initial noise: $x_T \sim \mathcal{N}(0, I)$
2  Initialize counter: $s = 1$
3  **for** $t \in [T - 1, \cdots, 0]$ **do**
4    **if** $\text{mod}(s, B) = 0$ **then**
5      Sample $N$ times over $B$ steps:
       $\{x_{t-1}^{(n)}\}_{n=1}^N \overset{i.i.d.}{\sim} \prod_{i=t}^{t+B} p(x_{i-1}|x_i)$
6      Compute values of all N samples:
       $\{V(x_{t-1}^{(n)})\}_{n=1}^N = \{r(\mathbb{E}[x_0|x_{t-1}^{(n)}])\}_{n=1}^N$
7      Select the sample with maximum value:
       $x_{t-1} \leftarrow \underset{\{x_{t-1}^{(n)}\}_{n=1}^N}{\text{argmax}} \ V(x_{t-1}^{(n)}; p, c)$
8    **end**
9    $s \leftarrow s + 1$
10 **end**
**Return:** $x_0$

---

value function estimation (line 6) using Eq. (12), select the best sample (line 7) to continue the reverse process until we obtain a clean image at $t = 0$. A key advantage of CoDe is its ability to achieve similar alignment-divergence trade-offs while using a significantly lower value of $N$, as is demonstrated in Section 5. A more detailed discussion on the optimal reward-tilted posterior in Eq. (10) and our blockwise controlled denoising procedure can be found in Appendix C.

**Best-of-N (BoN) sampling for diffusion models.** A strong baseline for inference-time alignment is Best-of-N (BoN). Empirical evidence from the realm of large language models (LLMs) (Gao et al., 2022; Mudgal et al., 2024; Gui et al., 2024) suggests that BoN closely approximates sampling from the optimal solution presented in Theorem 2.1, which is theoretically corroborated by Beirami et al. (2024); Yang et al. (2024). More recently, BoN has emerged as a strong baseline for scaling inference-time compute (Snell et al., 2024; Brown et al., 2024). In BoN, $N$ samples are obtained from the diffusion model by completely unrolling it out over $T$ denoising steps. Then, the most favorable image is selected based on a reward. This renders BoN sampling equivalent to CoDe with $B = T$. For other intermediate values of $B$, CoDe could be seen as a blockwise generalization of BoN.

**Soft Value-Based Decoding (SVDD) for diffusion models.** Concurrently to our work, Li et al. (2024) proposed an iterative sampling method to integrate soft value function-based reward guidance into the standard inference procedure of pre-trained diffusion models. The soft value function helps look-ahead into

how intermediate noisy states lead to high rewards in the future. Specifically, this method involves first sampling $N$ samples from the base diffusion model, and then selecting the sample corresponding to the highest reward across the entire set. This highest-reward sample is used for the next denoising step in the reverse-diffusion process. This renders SVDD-PM sampling as a special case of `CoDe`, operating specifically on a step block size $B = 1$.

**Why blockwise BoN is almost optimal** `CoDe`'s technique for sampling from the reward-tilted posterior and its theoretical optimality in terms of reward vs divergence tradeoffs has also been used in related works such as (Li et al., 2024; Beirami et al., 2024; Gui et al., 2024; Yang et al., 2024). Specifically on the optimality of this sampling technique, we would like to mention that several recent works have shown that BoN sampling is almost optimal in terms of reward vs divergence tradeoffs (Beirami et al., 2024; Gui et al., 2024; Yang et al., 2024; Mudgal et al., 2024). In particular, Theorem 1 from (Yang et al., 2024) shows that the samples obtained from BoN follow the same distribution as the optimal CD from Eq. (10). This is the reason (Mudgal et al., 2024) reported the most favorable reward vs divergence tradeoffs using blockwise language model decoding as blockwise decoding is also optimal in terms of reward vs divergence given that it interpolates two (almost) optimal decoding schemes.

## 4 Experimental Setup

We assess the performance of `CoDe` by comparing it against a suite of existing state-of-the-art guidance methods, in Text-to-Image (T2I) and Text-and-Image-to-Image ((T+I)2I) scenarios, across both differentiable and non-differentiable reward models. Unless otherwise mentioned, for all experiments, we use a pretrained Stable Diffusion version 1.5 (Rombach et al., 2021) as our base model, which is trained on the LAION-400M dataset (Schuhmann et al., 2021). As highlighted earlier, we strive to present meaningful comparative (both qualitative and quantitative) results across a variety of scenarios. For quantitative evaluations, we generate 50 images per setting (i.e., prompt-reference image pair) with 500 DDPM steps. To achieve this, we have used NVIDIA A100 GPUs with 80GB of RAM. Through extensive experiments, we aim to answer the following questions: *Does CoDe offer a comptetitive alignment-divergence trade-off compared to other baselines? How does CoDe perform across guidance tasks qualitatively and quantitatively?*

**Baselines.** We select a set of widely adopted baselines from the literature. Recall that our goal is to sample from the optimal value of the KL-regularized objective, as outlined in Theorem 2.1. One approach to achieve this, as detailed in Appendix B, is using a gradient-based method with an approximated value function, as in DPS (Chung et al., 2023), which serves as our first baseline. Further, Universal Guidance (UG) (Bansal et al., 2024b), MPGD (He et al., 2024) and Freedom (Yu et al., 2023), improve upon DPS by offering better gradient estimation. Another way to sample from Theorem 2.1 is by using a sampling-based approach such as in `CoDe`. In this direction, we consider Best-of-N (BoN) (Beirami et al., 2024) and SVDD-PM (Li et al., 2024) as our third and fourth baselines, which are also special cases of `CoDe` as explained earlier. For the sake of completeness, we also consider SDEdit (Meng et al., 2021) as a relevant (T+I)2I approach, for which all baselines could build on.

**Extensions with Noise Conditioning.** When the reward distribution deviates significantly from the base distribution $p$, sampling-based approaches would require a relatively larger value of $N$ to achieve alignment. To tackle this, a reference input sample, e.g. an image with the desired paining stroke or style, denoted as $x_{\mathrm{ref}}$, is provided as an additional conditioning input. Next to that, inspired by image editing techniques using diffusion (Meng et al., 2021; Koohpayegani et al., 2023), we add partial noise corresponding to only $\tau = \eta \times T$ (with $\eta \in (0, 1]$) steps of the forward diffusion process, instead of the full noise corresponding to $T$ steps. Then, starting from this noisy version of the reference image $x_\tau$, `CoDe` progressively denoises the sample for only $\tau$ steps to generate the clean, reference-aligned image $x_0$. By conditioning the initial noisy sample $x_\tau$ on the reference image $x_{\mathrm{ref}}$, we can generate images $x_0$ that better incorporate the characteristics and semantics of the reference image while adhering to the text prompt $c$. An extended version of Algorithm 1 with noise-conditioning, denoted as `CoDe`($\eta$) is discussed in detail in Appendix C (see lines $1 - 3$ in Algorithm 2). For the sake of fair comparison, we apply this enhancement also to other (T+I)2I baselines denoting them as BoN($\eta$), SVDD-PM($\eta$), UG ($\eta$) and DPS($\eta$). As we demonstrate in our experimentation, threshold $\eta$ provides an *extra knob* built in `CoDe` allowing the user to efficiently trade off divergence for reward. Note

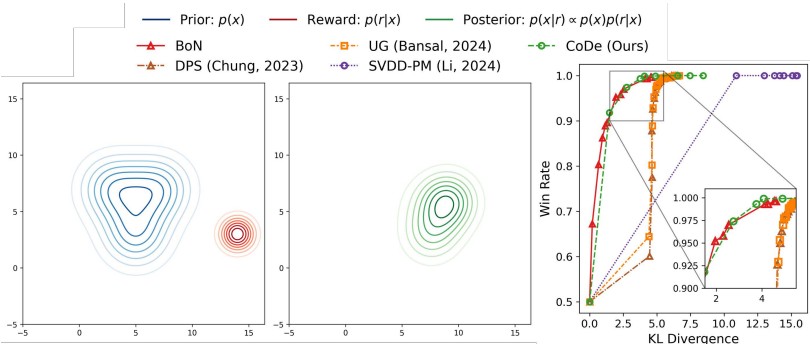

Figure 2: Setup (left, middle) and reward vs. divergence trade-off (right) for Case Study I. CoDe offers highest reward at lowest divergence with much lower $N$ than BoN.

that the reward-conditioning of the generated image is inversely proportional to the value of $\eta$. Setting $\eta = 1$ results in $\tau = T$ and fully deactivates the input-image conditioning. A byproduct of this conditioning is compute efficiency, as is discussed in Section 8.

**Evaluation Settings and Metrics.** We consider two case studies. **Case Study I**: a prototypical 2D Gaussian Mixture Models (GMMs) in Section 5, as is also studied in (Ho et al., 2021; Wu et al., 2024); **Case Study II**: widely adopted T2I and (T+I)2I evaluations using Stable Diffusion in Section 6 across five reward-alignment scenarios: (i) style, (ii) face (iii) stroke, (iv) compressibility guidance, and (v) aesthetic guidance. For Case Study I, we present trade-off curves for win rate versus KL-divergence for all baselines. For Case Study II, since calculating KL-divergence in high-dimensional image spaces is intractable, we use Frechet Inception Distance (FID) (Heusel et al., 2017). To ensure we capture alignment w.r.t reference image (and avoid using the guidance reward itself) we borrow an image alignment metric commonly used in style transfer domain (Gatys et al., 2016; Yeh et al., 2020), referred to as I-Gram here. Further, we assess prompt alignment using CLIPScore (Hessel et al., 2021), referred to as T-CLIP throughout the paper. Additionally, we consider win rate (commonly adopted in the LLM space) as yet another evaluation metric, where it reflects on the number of samples offering larger reward than the base model. To sum up, we consider expected reward, FID, I-Gram, T-CLIP, and win rate.

## 5 Case Study I: Gaussian Mixture Models (GMMs)

To establish an in-depth understanding of the impact of the proposed methods, we start with a simple model/reward distribution as shown in Fig. 2. For the prior distribution, we consider a 2D Gaussian mixture model $p(\boldsymbol{x}_0) = \sum_{i=0}^{2} w_i \mathcal{N}(\boldsymbol{\mu}_i, \sigma^2 \boldsymbol{I}_2)$, where $\sigma = 2$, $[\boldsymbol{\mu}_1, \boldsymbol{\mu}_2, \boldsymbol{\mu}_3] = [(5, 3), (3, 7), (7, 7)]$, and $\boldsymbol{I}_d$ is an $d$-dimensional identity matrix. Additionally, we define the reward distribution as $p(r|\boldsymbol{x}) = \mathcal{N}(\boldsymbol{\mu}_r, \sigma_r^2 \boldsymbol{I}_2)$ with $\boldsymbol{\mu}_r = [14, 3]$ and $\boldsymbol{\sigma}_r = 2$. As can be seen in the figure, in this case and by design, reward distribution is far off the peak of the prior. Here, we train a diffusion model with a 3-layer MLP that takes as input $(\boldsymbol{x}_t, t)$ and predicts the noise $\boldsymbol{\varepsilon}_t$. This model is trained over 200 epochs with $T = 1000$ denoising steps. Note that all other discussed baselines can straightforwardly be trained in this setting. The results are illustrated in Fig. 2 where we plot win rate vs. KL-divergence for different values of $N \in [2, 500]$. The details for computing the KL divergence have been provided in the appendix F. For the guidance-based methods DPS and UG, the guidance scale is varied between 1 and 50, whereas for the sampling-based methods, BoN the number of samples $N$ is varied between 2 and 500, while for SVDD and CoDe, the number of samples $N$ is varied between 2 and 40.

As can be seen, BoN offers the upper bound of performance with CoDe achieving on-par performance trade-offs. This aligns with the observations from the realm of LLMs (Beirami et al., 2024; Gui et al., 2024), where BoN has been theoretically proven to offer the best win-rate vs KL divergence trade-offs. However, it is important to notice that CoDe achieves an on-par win-rate vs KL divergence trade-off with BoN for a much smaller $N$. Specifically, CoDe with $N \in [2, 10]$ achieves the same win rate vs KL divergence performance as BoN with $N \in [30, 500]$, rendering CoDe roughly 10-15× more efficient than BoN.

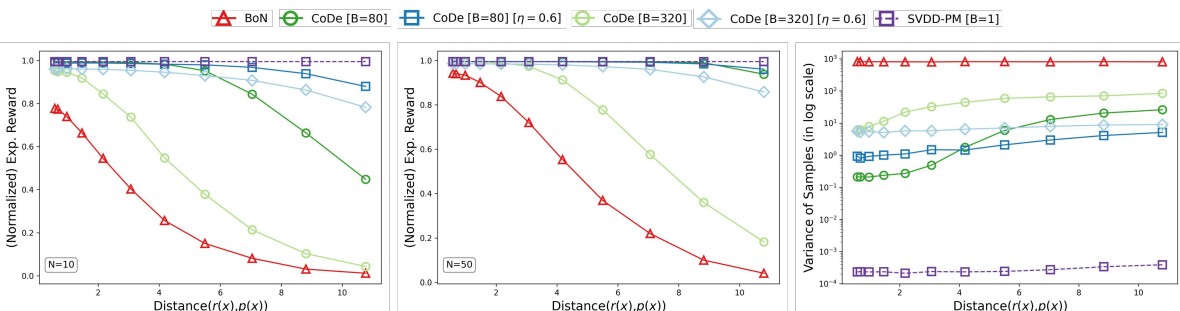

Figure 3: In contrast to BoN, SVDD-PM, `CoDe` with and without noise-conditioning ($\eta = 0.6$, $\eta = 1$, resp.) are robust against increased distance between reward and prior distributions. SVDD-PM's generated samples offer almost zero variance indicating reward over-optimization.

In contrast, UG and DPS tend to exhibit higher KL divergence, as they often collapse to the mode of the reward distribution when the guidance scale is increased, leading to a reduction in diversity among the sampled data points, a phenomenon also noted in prior research (Sadat et al., 2024; Ho et al., 2021). In both scenarios, SVDD achieves a high expected reward (or win rate) but at the expense of significantly higher divergence, even for $N = 2$. In contrast, `CoDe` offers flexibility, allowing users to balance the trade-off by adjusting parameters such as $N$ and $B$, as is demonstrated here. For a different scenario (and for providing a more comprehensive picture), where the reward distribution falls within the distribution of the prior see Appendix D.

Let us dive one step deeper into comparing the performance of `CoDe`, `CoDe`($\eta$), BoN and SVDD-PM. To this aim, in Fig. 3, we vary the distance between the mean of the reward and prior distributions, gradually shifting the reward further away. To handle this scenario effectively, we use noise conditioning for `CoDe`, denoted by ($\eta = 1$, 0.6), by sampling from the *known* reward distribution and providing it as an input conditioning sample. We also study the impact of block size $B$ for reward-guidance by varying it between $B = [1, 80, 320]$, with $B = 1$ corresponding to SVDD-PM. This is shown for $N = 10, 50$ in Fig. 3 where the expected reward sharply drops for BoN regardless of choice of $N$, whereas it drops less or remains almost intact for `CoDe` with $\eta = 0.6$, $B = [80, 320]$. In the case of $\eta = 1$ for `CoDe`, we notice that the reward drops sharply for a larger block size ($B = 320$), while almost remaining constant or dropping lesser for a smaller block size ($B = 80$). On the other hand, SVDD-PM, imposing token-wise aggressive guidance with $B = 1$ offers a high, constant reward for both $N = [10, 50]$. However, SVDD-PM's generated samples have a variance that is orders of magnitude lower than BoN or `CoDe`, as can be seen in the right most part of Fig. 3. This particularly low variance of SVDD-PM's generated samples (almost $10^{-4}$) indicates their collapse to a single point in the reward distribution. This has been studied extensively in the literature and is referred to as reward over-optimization (Prabhudesai et al., 2023), and corroborates the need for keeping a small KL divergence from the base model, as also empirically and theoretically argued by (Beirami et al., 2024; Gao et al., 2022)

## 6 Case Study II: Image Generation with Stable Diffusion

We consider five commonly adopted guidance scenarios: *compressibility*, *style*, *stroke*, *face* and *aesthetics* guidance. We consider T2I tasks for compressibility and aesthetics guidance scenarios and (T+I)2I tasks for compressibility, style, face and stroke guidance scenarios. For each scenario, the reward model is task specific as elaborated in the following. A text prompt as well as a reference image are used as guidance signals. For the first three scenarios, a total of 33 generation settings (i.e., text prompt - reference image pairs) are used for evaluations. For compressibility guidance, we have 12 settings. Per setting, we generate 50 samples and estimate the evaluation metrics accordingly. On the qualitative side, to demonstrate the capacity of `CoDe`($\eta$) compared to other baselines, we illustrate a few generated examples across two reference images for two different text prompts. In favor of space, the qualitative results for face and tradeoff curves for aesthetics guidance are deferred to the Appendix E, Figs. 15, 16 and Fig. 8, respectively. On the quantitative side, given the non-differentiable nature of compressibility as guidance signal, we demonstrate the efficacy of `CoDe` as compared to only sampling-based baselines in Table 1. For differentiable reward-guidance scenarios (style, face and stroke), we evaluate the performance across all scenarios/settings combined for further statistical significance in Tables 2, 4.

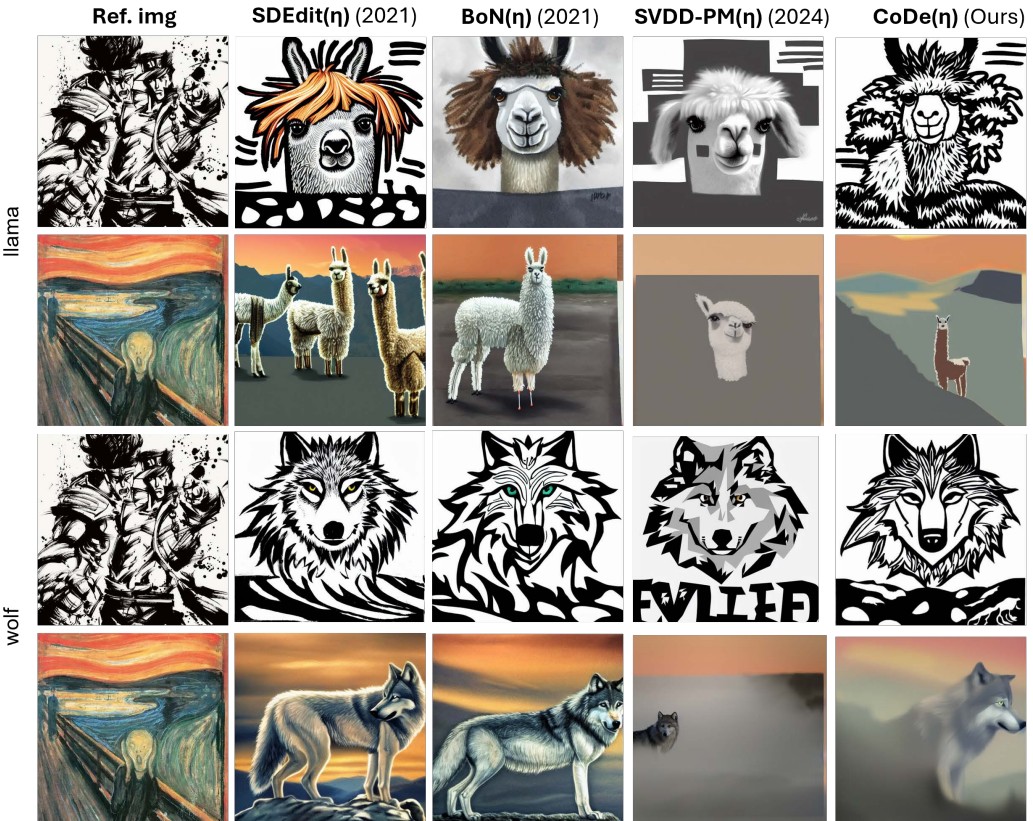

Figure 4: CoDe($\eta$) demonstrates a superior trade-off between compressibility, image and text alignment as compared to other baselines on the (T+I)2I settings.

## 6.1 Non-Differentiable Reward: Compression

First, we consider a scenario with non-differentiable reward, where gradient-based guidance does not apply. Following Fan et al. (2023), we use image compressibility as the reward score which is measured by the size of the JPEG image in kilobytes. This way, we guide the diffusion denoising process to generate memory-light, compressible images.

**Qualitative Comparisons.** A comparative look across baselines and settings is illustrated in Fig. 4. We observe that CoDe($\eta$) generates the best results, offering superior compression as well as image and text alignment. SVDD-PM($\eta$), SDEdit($\eta$) and BoN($\eta$) align well with the image and text prompt, but fall short on providing smooth-textured, content-light compressed images. However, this is to be expected in the case of SDEdit($\eta$) since its generative process is not guided by the compression-reward.

**Quantitative Evaluations.** Table 1 illustrates the performance comparison of CoDe, CoDe($\eta$) as compared to other baselines. Sampling-based baselines

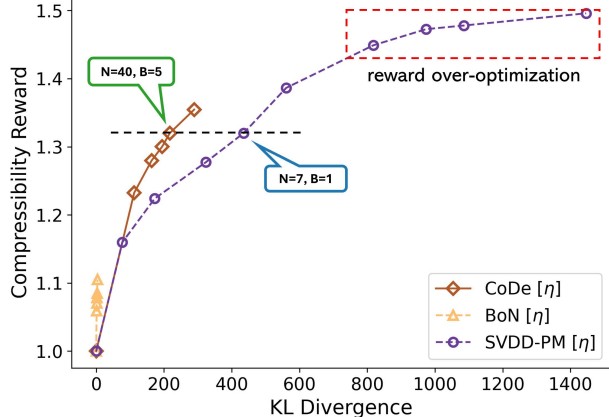

Figure 5: CoDe($\eta$) offers a better reward vs. KL-divergence trade-off as compared to BoN($\eta$) for all N values. SVDD-PM($\eta$) demonstrates a higher reward beyond $N = 7$, but at the cost of a much higher KL-divergence.

(SVDD-PM Li et al. (2024) and BoN Gao et al. (2022)) for two scenarios, T2I and (T+I)2I, where in the latter the reference image is omitted. In both scenarios, we observe that SVDD-PM and SVDD-PM($\eta$) offer

slightly higher compression reward score as compared to other baselines; however, CoDe($\eta$) offers better image (I-Gram) and text (T-CLIP) alignment and the least divergence from the base distribution (FID, CMMD) as compared to all other baselines. Most notably, Fig. 5 illustrates the reward vs. KL divergence for this scenario, demonstrating that in normal operating regimes (before reward over-optimization occurs, see appendix I, Fig. 19), CoDe($\eta$) offers almost the same reward as its special case of $B = 1$ for SVDD-PM($\eta$) with less than half of its KL divergence. Here, different points on the curves represent sweeping on each method's main set of parameters ($N = [10, 20, 30, 40, 100]$ for CoDe($\eta$), BoN($\eta$) and $N = [2, 3, 5, 7, 10, 20, 30, 40, 100]$ for SVDD-PM($\eta$)). Details on the computation of KL divergence are mentioned in the appendix section F.

## 6.2 Differentiable Rewards

**Style guidance.** We guide image generation based on a reference style image (Bansal et al., 2024b; He et al., 2024; Yu et al., 2023). Following the reward model proposed in Bansal et al. (2024b), we use the CLIP image encoder to obtain embeddings for the reference style and the generated images. The cosine similarity between these embeddings is then used as the guidance signal. **Stroke guidance.** A closely related scenario to style guidance is stroke

Table 1: Quantitative metrics for compression reward.

| Method | Compressibility Reward - T2I | | | | |
|---|---|---|---|---|---|
| | Rew. (↑) | FID (↓) | CMMD (↓) | T-CLIP (↑) | I-Gram (↑) |
| Base-SD | 1.0 | 1.0 | 1.0 | 1.0 | - |
| BoN | 1.23 | 1.10 | 1.70 | 0.99 | - |
| SVDD-PM | 1.83 | 2.86 | 61.75 | 0.88 | - |
| CoDe | 1.65 | 2.12 | 32.70 | 0.95 | - |
| | Compressibility Reward - (T+I)2I | | | | |
| Base-SD | 1.0 | 1.0 | 1.0 | 1.0 | 1.0 |
| SDEdit ($\eta = 0.8$) | 0.97 | 2.19 | 29.25 | 0.98 | 1.34 |
| BoN ($\eta = 0.8$) | 1.08 | 2.27 | 31 | 0.98 | 1.32 |
| SVDD-PM ($\eta = 0.8$) | 1.48 | 3.54 | 69.5 | 0.89 | 1.15 |
| CoDe($\eta = 0.8$) | 1.34 | 3.08 | 48.75 | 0.97 | 1.20 |

generation, where a high-level reference image containing only coarse colored strokes is used as reference (Cheng et al., 2023; Meng et al., 2021). The objective in this setting is to produce images that remain *faithful* to the reference strokes. To achieve this, similar to style guidance, we employ the CLIP image encoder to obtain embeddings from both the reference and generated images and compute the reward by measuring the cosine similarity between these embeddings. **Face guidance.** To guide the generation process to capture the face of a specific individual (as in (He et al., 2024; Bansal et al., 2024b)), we employ a combination of multi-task cascaded convolutional network (MTCNN) (Zhang et al., 2016) for face detection and FaceNet (Schroff et al., 2015) for facial recognition, which together produce embeddings for the facial attributes of the image. The reward is then computed as the negative $\ell_1$ loss between feature embeddings of the reference and generated images.

**Qualitative Comparisons.** A comparative look across baselines, scenarios and settings is illustrated in in Figs. 6 and 7 (and 12 15, 16 and 17 in the appendix). Let us start with style guidance in Fig. 6. As can be seen, CoDe($\eta$) shows arguably a better performance in capturing the style of the reference image, regardless of the text prompt. When it comes to alignment

Table 2: Quant. metrics (± std.) for (T+I)2I differentiable scenarios.

| Method | FID (↓) | I-Gram (↑) | T-CLIP (↑) | Runtime (↓) |
|---|---|---|---|---|
| SDEdit($\eta$) | 1.0 | 1.0 | 1.0 | 1.0 |
| BoN($\eta$) | 1.06 | 1.08 (± 0.002) | 0.98 (± 0.002) | 23.62 (± 0.005) |
| SVDD-PM ($\eta$) | 1.29 | 1.64 (± 0.03) | 0.94 (± 0.002) | 103.73 (± 0.05) |
| DPS ($\eta$) | 1.01 | 1.23 (± 0.04) | 0.96 (± 0.005) | 6.07 (± 0.03) |
| UG($\eta$) | 1.38 | 1.31 (± 0.05) | 0.89 (± 0.002) | 92.07 (± 0.04) |
| CoDe($\eta$) | 1.15 | 1.60 (± 0.05) | 0.98 (± 0.006) | 37.21 (± 0.03) |

with the text prompt, UG($\eta$) seems to suffer to some extent with "Eiffel tower" and "woman" fading away in the corresponding images.

**Important Remark:** Note that by excluding noise conditioning from the original baselines (removing $\eta$, see Figs. 12, 22), they all suffer in capturing the style of the reference image, highlighting the importance of using noise-conditioning as is proposed for CoDe for all baselines operating in the (T+I)2I scenarios. Further qualitative results for stroke guidance are summarized in Fig. 7, where the same narrative and observations extend. The results for face guidance are deferred to the Appendix E, Figs. 15 and 16.

**Quantitative Evaluations.** Table 2 summarizes the performance across all scenarios (including all settings) over four metrics: I-Gram, FID, T-CLIP and runtime (in second/image, and detailed Section 6.4). The reason why we use I-Gram (instead of expected reward per scenario) in our evaluations is because expected reward has been *seen* by the model throughout the guidance process.

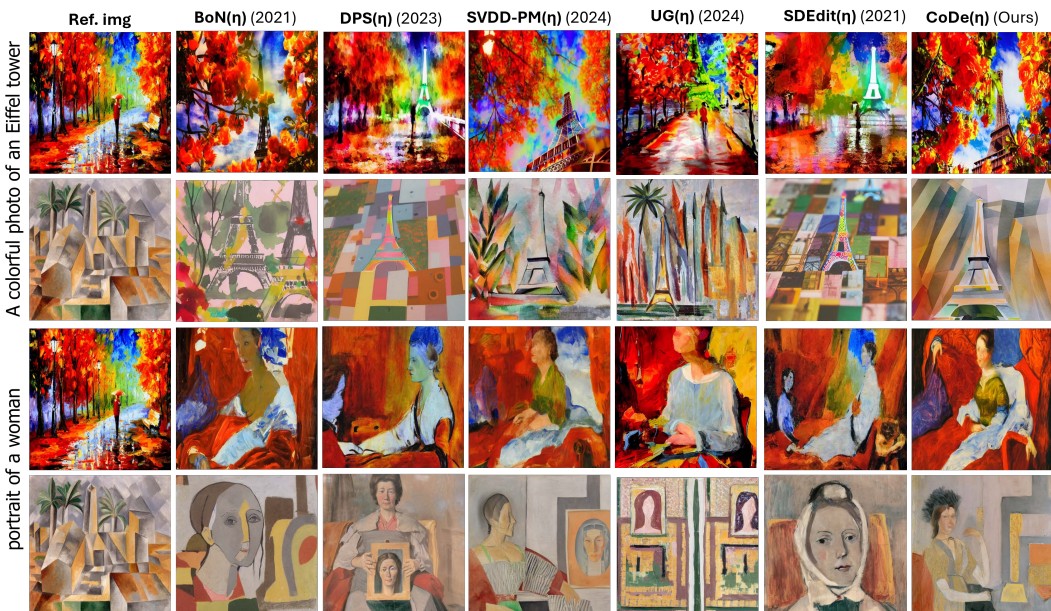

Figure 6: The style alignment offered by CoDe($\eta$) stands on par or outperforms other baselines in terms of quality and preserving nuances of the reference image, while adhering to the text-prompt.

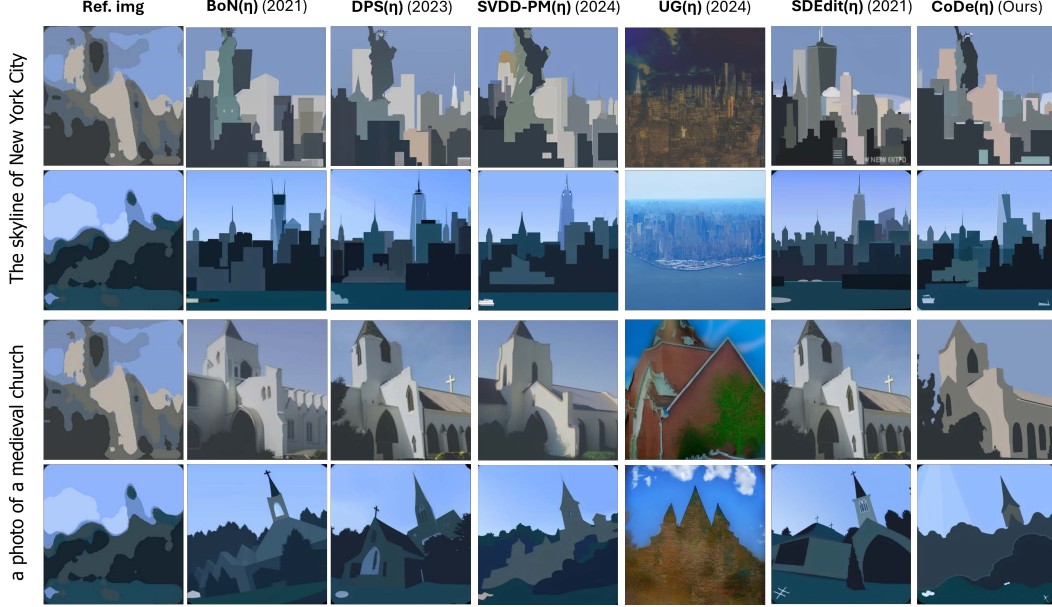

Figure 7: Same narrative as in Fig. 6 with CoDe outperforming UG($\eta$) in terms of quality and ref. image-alignment, while standing-on par with all other baselines.

The scores here are normalized with respect to SDEdit as the baseline, thus indicating the performance gain over that. We notice that SVDD-PM($\eta$) and CoDe($\eta$) perform on par in terms of offering the best image alignment (indicated by I-Gram), while being superior than all other baselines. However, CoDe($\eta$) offers a better trade-off between image, text-alignment and divergence as compared to SVDD-PM($\eta$), as indicated by its superior T-CLIP and FID scores. Note that here again by excluding noise-

Table 3: Quant. metrics ($\pm$ std.) for aesthetics guidance.

| Method | Aesthetic Guidance - T2I | | | |
|---|---|---|---|---|
| | **Rew.** ($\uparrow$) | **FID** ($\downarrow$) | **CMMD** ($\downarrow$) | **T-CLIP** ($\uparrow$) |
| Base-SD (2021) | 1.0 | 1.0 | 1.0 | 1.0 |
| BoN (2022) | 1.10 | 1.98 | 6.41 | 0.99 |
| UG (2024b) | 1.30 | 7.53 | 65.05 | 0.86 |
| MPGD (2024) | 1.22 | 6.55 | 57.63 | 0.93 |
| Freedom (2023) | 1.29 | 4.07 | 22.45 | 0.95 |
| **CoDe** | 1.27 | 2.59 | 6.6 | 0.99 |

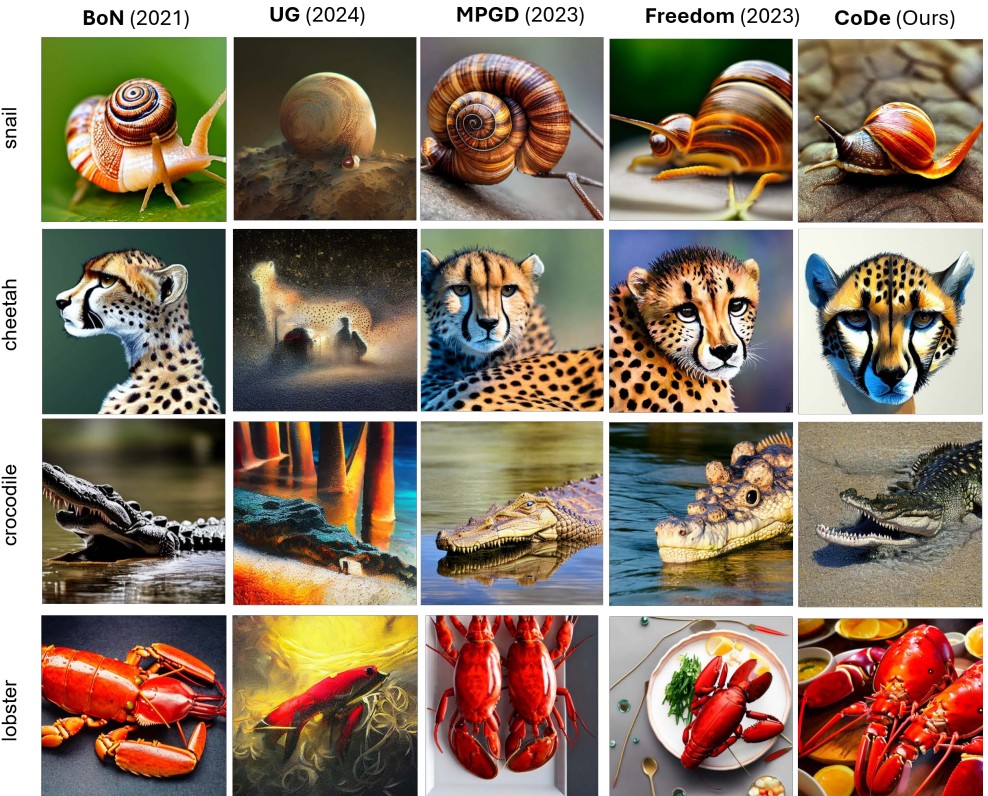

Figure 8: Qualitative evaluation across methods for aesthetic guidance.

conditioning from the other baselines (as in their original proposition) the gain margin offered by CoDe($\eta$) would be considerably larger as is shown in our ablation studies. See Appendix E for further qualitative and quantitative results.

**Aesthetic Guidance**. To guide the diffusion denoising process towards generating aesthetically pleasing images, we employ the LAION aesthetic predictor V2 (Schuhmann et al., 2022), which leverages a multi-layer perceptron (MLP) architecture trained atop CLIP embeddings. This model's training data consists of 176,000 human image ratings, spanning a range from 1 to 10, with images achieving a score of 10 being considered art piece. Table 3 shows the results for sampling based and gradient based inference-time guidance methods on the given T2I scenario. We observe that CoDe offers better rewards as compared to MPGD (He et al., 2024) and BoN while being second to best as compared to Freedom Yu et al. (2023) and UG Bansal et al. (2024b). However, CoDe offers better text alignment (T-CLIP) and lower divergence from the base distribution (FID, CMMD) as compared to all its gradient based counterparts. This can also be observed in Figs. 18, where CoDe offers almost the same reward as MPGD, Freedom and UG, but at a lower divergence or higher T-CLIP. Additionally, we demonstrate a qualitative comparsion between all baselines in Fig. 8. It can be observed that UG generates aesthetic images that do not completely adhere to the text-prompt leading to reward over-optimization. This is not as prominent in Freedom, CoDe and MPGD where the generated images are of comparable aesthetic quality while significantly adhering to the text prompt of the animal.

## 6.3   Ablations

Fig. 9 investigates the impact of varying block size ($B$) and noise ratio ($\eta$) for CoDe on image (I-Gram) vs. text alignment (T-CLIP). For reference, CoDe($\eta = 1$) (without image-conditioning) and UG are also depicted. Here, different points per curve represent sweeping on their main parameter ($N = [5, 10, 20, 30, 40, 100]$ for CoDe, and guidance scale of $[1, 3, 6, 12, 24]$ for UG). On the left image, increasing block size seems to limit the image alignment performance; or put differently, same performance at a much larger $N$. Regardless of block size, CoDe curves fall on top of UG indicating a superior overall performance. On

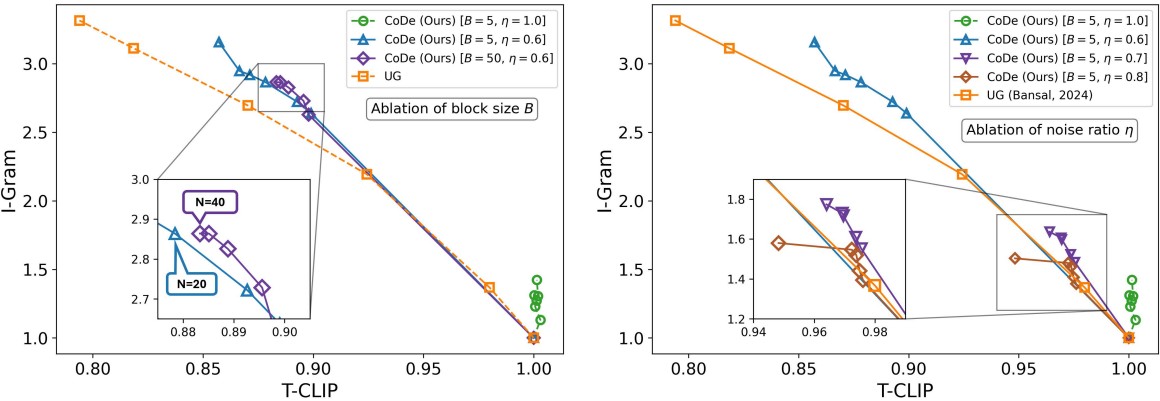

Figure 9: Ablation on the block size ($B$) and the noise ratio ($\eta$).

the right, changing the noise ratio $\eta$ toward higher values, reduces the conditioning strength (as indicated also in (Meng et al., 2021; Koohpayegani et al., 2023)) resulting in lower image alignment capacity (I-Gram). Yet again, CoDe variants fall on top of the UG curve suggesting better image vs. text alignment performance. More detailed ablation studies and reward vs alignment trade-off curves are provided in Appendix E. Further note that the operation points with very low T-CLIP scores on UG curves ended up degenerating to the extent that images did not have anything in common with the text prompt (see appendix J, Fig. 20), which was another consideration for choosing the best trade-off point.

We also study the impact of dropping the partial-noise conditioning on all baselines, including CoDe in Table 4. For reference, CoDe($\eta$) is also included where the best empirical value for $\eta$ is selected per scenario. We report scores across all metrics by normalizing them w.r.t. the base Stable Diffusion model (denoted by Base-SD). As can be seen, CoDe, i.e. without noise-conditioning, offers performance gains in terms of image alignment while staying

Table 4: Ablation on partial-noise conditioning.

| Method | FID ($\downarrow$) | I-Gram ($\uparrow$) | T-CLIP ($\uparrow$) | Runtime ($\downarrow$) |
|---|---|---|---|---|
| Base-SD (2021) | 1.0 | 1.0 | 1.0 | 1.0 |
| BoN (2022) | 1.19 | 1.07 ($\pm$ 0.004) | 0.99 ($\pm$ 0.001) | 18.90 ($\pm$ 0.01) |
| SVDD-PM (2024) | 1.42 | 1.24 ($\pm$ 0.02) | 0.98 ($\pm$ 0.004) | 99.10 ($\pm$ 0.08) |
| DPS (2023) | 1.14 | 1.12 ($\pm$ 0.01) | 0.98 ($\pm$ 0.004) | 5.82 ($\pm$ 0.02) |
| UG (2024b) | 2.91 | 1.86 ($\pm$ 0.03) | 0.85 ($\pm$ 0.005) | 87.92 ($\pm$ 0.03) |
| CoDe | 1.17 | 1.30 ($\pm$ 0.009) | 0.99 ($\pm$ 0.001) | 34.63 ($\pm$ 0.04) |
| CoDe($\eta$) | 3.00 | 3.19 ($\pm$ 0.05) | 0.87 ($\pm$ 0.006) | 23.82 ($\pm$ 0.03) |

competitive w.r.t. text alignment (I-Gram and T-CLIP scores) and deviating lesser from the base model (FID score), compared to all baselines except UG. Notably, CoDe is also considerably faster than both SVDD-PM and UG in terms of runtime. As stated earlier, here CoDe($\eta$), i.e. with noise-conditioning, offers a much more pronounced gain in terms of I-Gram in terms of the other baselines. We provide general guidelines on setting $N, B, \eta$ for CoDe in Appendix G and an ablation on using adaptive guidance control for $N, B$ in Appendix H.

## 6.4 Computational Complexity.

We provide a comparative look at the complexity of the proposed approach against other baselines. To this aim, we consider two aspects: (i) the number of inference steps, (ii) the number of queries to the reward model. We then measure the overall runtime complexity in terms of time (in sec.) required to generate one image. This is summarized in Table 5. From a runtime perspective, within the gradient-based guidance group, DPS is relatively faster across all three generation scenarios. This is due to the $m$ gradient and $K$ refinement steps used in UG, which

Table 5: Computational complexity.

| Methods | Inf. Steps | Rew. Queries | Runtime [sec/img] |
|---|---|---|---|
| Base-SD (2021) | $T$ | - | 14.12 |
| BoN (2022) | $NT$ | $N$ | 266.77 |
| SVDD-PM (2024) | $NT$ | $NT$ | 1399.36 |
| DPS (2023) | $T$ | $T$ | 82.19 |
| UG (2024b) | $mKT$ | $mKT$ | 1241.47 |
| CoDe | $NT$ | $NT/B$ | 489.00 |
| CoDe($\eta$) | $N\eta T$ | $N\eta T/B$ | 336.39 |

are not used in DPS. Within the sampling based group, SVDD-PM, imposing token-wise aggressive guidance ($B = 1$), turns out to be an order of magnitude slower than BoN. CoDe, CoDe($\eta$) with its blockwise guidance remains to be faster and more efficient than BoN as well as UG, offering a $4\times$ faster runtime than UG.

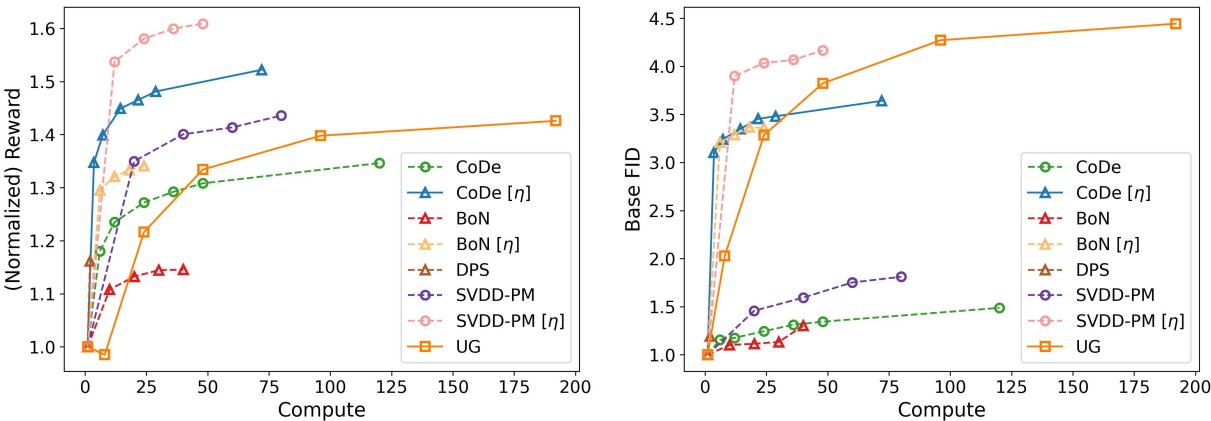

Figure 10: Reward vs. Compute and FID vs Compute trade-off curves for Style guidance.

## 6.5   Performance vs Efficiency Tradeoffs.

In addition to the breakdown of computational complexity, we also illustrate performance-efficiency tradeoff curves in terms of reward vs compute and divergence (FID) vs compute curves for the style guidance scenario in Fig. 10. Compute is calculated using the breakdown of the computational complexity in terms of the inference steps and reward queries as shown in Table 5. We illustrate both these curves since it is important to analyze both reward and divergence (FID) to get a holistic picture of performance and reward over-optimization. Here different points on the curve represent sweeping on their main parameters ($N = [5, 10, 20, 30, 40, 100]$ for CoDe, $N = [10, 20, 30, 40]$ for SVDD-PM, BoN, gradient guidance scale = $[0.5, 0.7, 0.9, 1.1, 1.3]$) for DPS and $K = [1, 3, 6, 12, 24]$ for UG (with the best gradient scale = 6). As can be observed, CoDe($\eta$) and SVDD-PM($\eta$) offer the best reward vs compute and FID vs compute tradeoffs as compared to all other baselines. Specifically, while SVDD-PM($\eta$) achieves higher rewards for the same compute as compared to CoDe($\eta$), it also deviates significantly more from the base distribution as compared to CoDe($\eta$). It is important to note that divergence captures preserving core capabilities not captured by reward, resulting in inferior reward vs divergence tradeoffs that are discussed in the previous sections. Thus, in terms of a tradeoff between performance (captured through reward and divergence) vs efficiency, CoDe($\eta$) still offers a better tradeoff as compared to SVDD-PM($\eta$) enabling performance points that are not even achievable by SVDD-PM($\eta$). On the other hand, UG offers high rewards but at the cost of either significantly higher compute or FID.

## 7   Related Work

**Finetuning-based alignment.** Prominent methods in this category typically involve either training a diffusion model to incorporate additional inputs such as category labels, segmentation maps, or reference images (Ho et al., 2021; Li et al., 2023; Zhang et al., 2023; Bansal et al., 2024a; Mou et al., 2024; Ruiz et al., 2023) or applying reinforcement learning (RL) to finetune a pretrained diffusion model to optimize for a downstream reward function (Prabhudesai et al., 2023; Fan et al., 2023; Wallace et al., 2023; Black et al., 2023; Gu et al., 2024; Lee et al., 2024; Uehara et al., 2024b). While these approaches have been successfully employed to satisfy diverse constraints, they are computationally expensive. Furthermore, finetuning diffusion models is prone to "reward hacking" or "over-optimization" (Clark et al., 2024; Jena et al., 2024), where the model loses diversity and collapses to generate samples that achieve very high rewards. This is often due to a mismatch between the intended behavior and what the reward model actually captures. In practice, a perfect reward model is extremely difficult to design. As such, here we focus on inference-time guidance-based alignment approaches where these issues can be circumvented. Additionally, none of the fine-tuning based methods are built for image-to-image scenarios, which is the focus of this work, as we clarified earlier. To compare against them, a direct approach could be fine-tuning per reference image, which renders the process

computationally infeasible, or taking a meta-learning approach to fine-tuning. However, such fundamental adjustments are beyond the current scope of our work.

**Gradient-based inference-time alignment.** There are two main divides within this category: (i) guidance based on a *value* function, and (ii) guidance based on a downstream *reward* function. In the first divide, a value function is trained offline using the noisy intermediate samples from the diffusion model. Then, during inference, gradients from the value function serve as signals to guide the generation process (Dhariwal & Nichol, 2021; Yuan et al., 2023). A key limitation of such an approach is that the value functions are specific to the reward model and the noise scales used in the pretraining stage. Thus, the value function has to be retrained for different reward and base diffusion models. The second divide of methods successfully overcomes this by directly using the gradients of the reward function based on the approximation of fully denoised images using Tweedie's formula (Chung et al., 2022; 2023; Yu et al., 2023). Interesting follow-up research has explored methods to reduce estimation bias (Zhu et al., 2023; Bansal et al., 2024b; He et al., 2024) and to scale gradients for maintaining the latent structures learned by diffusion models (Guo et al., 2024). Despite such advancements, the need for differentiable guidance functions can limit the broader applicability of the gradient-based methods.

**Gradient-free inference-time alignment.** Tree-search alignment has recently gained attention in the context of autoregressive language models (LMs), where it has been demonstrated that Best-of-$N$ (BoN) approximates sampling from a KL-regularized objective, similar to those used in reinforcement learning (RL)-based finetuning methods (Gui et al., 2024; Beirami et al., 2024; Gao et al., 2022). This approach facilitates the generation of high-reward samples while maintaining closeness to the base model. Mudgal et al. (2024) demonstrate that the gap between Best-of-$N$ (BoN) and token-wise *value-based* decoding (Yang & Klein, 2021) can be bridged using a blockwise decoding strategy. Inspired by this line of research, we propose a simple blockwise alignment technique (tree search with a fixed depth) that offers key advantages: (i) it preserves latent structures learned by diffusion models without requiring explicit scaling adjustments, unlike gradient-based methods, and (ii) it avoids "reward hacking" typically associated with learning-based approaches. Concurrently, Li et al. (2024) propose a related method, called SVDD-PM, based on the well-known token-wise decoding strategy in the LM space. In contrast, we devise a blockwise sampling strategy because it allows further control on the level of intervention, and offers a trade-off between divergence and alignment, which is of primal interest in the context of guided generation. To enhance the sampling strategy in terms of efficiency, we apply adjustable noise-conditioning which also offers greater control over guidance signals and further improves alignment. Sequential Monte Carlo-based methods (SMC) for diffusion models (Wu et al., 2023; Chung et al., 2023; Phillips et al., 2024; Cardoso et al., 2023) share similarities with tree-search-based alignment methods such as ours, particularly in not requiring differentiable reward models. However, these methods were originally designed to solve conditioning problems rather than reward maximization. Crucially, they involve resampling across an entire batch of images, which can lead to suboptimal performance when batch sizes are small since the SMC theoretical guarantees hold primarily with large batch sizes. In contrast, our method performs sampling on a per-sample basis. Lastly, using SMC for reward maximization can also result in a loss of diversity, even with large batch sizes.

## 8 Concluding Remarks

We introduce a gradient-free blockwise inference-time guidance approach for diffusion models. By combining blockwise optimal sampling with an adjustable noise conditioning strategy, `CoDe`, `CoDe`($\eta$) offer a better reward vs. divergence trade-off compared to state-of-the-art baselines.

**Limitations and future work.** Diffusion models are still computationally intensive; as such, extracting quantitative results on the performance of (inference-time) guidance-based alignment methods calls for massive resources, especially when ablating across numerous design parameters. We have used up to 32 NVIDIA A100's solely dedicated to the presented evaluation results. Yet, most commonly adopted settings we have experimented with to arrive at the numerical results in Tables 1 and 4 can be further expanded for the sake of better statistical significance in future work.

**Broader Impact.** We would like to caution against the blind usage of the proposed techniques as alignment methods are prone to reward over-optimization, which warrants care in socially consequential applications.

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

## A Proof of Theorem 2.1

*Proof of Theorem 2.1.*

$$J_\lambda(x_t, \pi, c) = \mathbb{E}_{x_{t-1} \sim \pi} \left[ \lambda(V(x_{t-1}; p, c) - V(x_t; p, c)) + \log \frac{p(x_{t-1}|x_t, c)}{\pi(x_{t-1}|x_t, c)} \right] \tag{13}$$

$$= \mathbb{E}_{x_{t-1} \sim \pi} \left[ \log \frac{p(x_{t-1}|x_t, c) \, e^{\lambda(V(x_{t-1}; p, c) - V(x_t; p, c))}}{\pi(x_{t-1}|x_t, c)} \right] \tag{14}$$

$$= \mathbb{E}_{x_{t-1} \sim \pi} \left[ \log \frac{p(x_{t-1}|x_t, c) \, e^{\lambda V(x_{t-1}; p, c)}}{\pi(x_{t-1}|x_t, c)} + \log e^{\lambda V(x_t; p, c)} \right] \tag{15}$$

$$= \mathbb{E}_{x_{t-1} \sim \pi} \left[ \log \frac{p(x_{t-1}|x_t, c) \, e^{\lambda V(x_{t-1}; p, c)}}{\pi(x_{t-1}|x_t, c)} \right] + \lambda V(x_t; p, c) \tag{16}$$

Now, let

$$p_\lambda(x_{t-1}|x_t, c) := \frac{p(x_{t-1}|x_t, c) e^{\lambda V(x_{t-1}; p, c)}}{Z_\lambda(x_t, c)}, \tag{17}$$

where the normalizing constant $Z_\lambda(x_t, c)$ is given by

$$Z_\lambda(x_t, c) = \mathbb{E}_{x_{t-1} \sim p} \left[ p(x_{t-1}|x_t, c) e^{\lambda V(x_{t-1}; p, c)} \right]. \tag{18}$$

Putting it back in Eq. 16, we get

$$J_\lambda(x_t, \pi, c) = \mathbb{E}_{x_{t-1} \sim \pi} \left[ \log \frac{p_\lambda(x_{t-1}|x_t, c)}{\pi(x_{t-1}|x_t, c)} Z_\lambda(x_t, c) \right] + \lambda V(x_t; p, c) \tag{19}$$

$$= \mathbb{E}_{x_{t-1} \sim \pi} \left[ \log \frac{p_\lambda(x_{t-1}|x_t, c)}{\pi(x_{t-1}|x_t, c)} + \log Z_\lambda(x_t, c) \right] + \lambda V(x_t; p, c) \tag{20}$$

$$= \mathbb{E}_{x_{t-1} \sim \pi} \left[ \log \frac{p_\lambda(x_{t-1}|x_t, c)}{\pi(x_{t-1}|x_t, c)} \right] + \log Z_\lambda(x_t, c) + \lambda V(x_t; p, c) \tag{21}$$

$$= -\mathbb{E}_{x_{t-1} \sim \pi} \left[ \log \frac{\pi(x_{t-1}|x_t, c)}{p_\lambda(x_{t-1}|x_t, c)} \right] + \log Z_\lambda(x_t, c) + \lambda V(x_t; p, c) \tag{22}$$

$$= -KL(\pi(x_{t-1}|x_t, c) \| p_\lambda(x_{t-1}|x_t, c)) + \log Z_\lambda(x_t, c) + \lambda V(x_t; p, c) \tag{23}$$

Eq. 23 is uniquely maximized by $\pi_\lambda^*(x_{t-1}|x_t, c) = p_\lambda(x_{t-1}|x_t, c)$. $\qquad \square$

# B  Sampling from Optimal Model using Langevin Dynamics

Given the optimal policy given in Eq. 10, our goal is to now sample from $\pi^*$ instead of $p$. However, given only $p$, it is difficult to sample from this optimal policy. To overcome this problem, we look at the score-based sampling approach as in NCSN (Song & Ermon, 2019). Starting from an arbitrary point $x_T$, we iteratively move in the direction of $\nabla_{x_t} \log \pi^*(x_t)$, which is equivalent to $\nabla_{x_t} \log p_\lambda(x_t)$. We can derive an equivalent form:

$$p_\lambda(x_t) = \frac{p(x_t)e^{\lambda V(x_t)}}{Z_\lambda} \tag{24}$$

$$\log p_\lambda(x_t) = \log p(x_t) + \lambda V(x_t) - \log Z_\lambda \tag{25}$$

$$\nabla_{x_t} \log p_\lambda(x_t) = \nabla_{x_t} \log p(x_t) + \nabla_{x_t} \lambda V(x_t) - \nabla_{x_t} \log Z_\lambda \tag{26}$$

$$s_\lambda(x_t, t) = s_\theta(x_t, t) + \lambda \nabla_{x_t} V(x_t). \tag{27}$$

As the above derivation is limited to stochastic diffusion sampling, we leverage the connection between diffusion models and score matching (Song & Ermon, 2019):

$$\nabla_{x_t} \log p(x_t) = -\frac{1}{\sqrt{1 - \bar{\alpha}_t}} \varepsilon_t. \tag{28}$$

**Similarity with classifier guidance**. Starting from an arbitrary point $x_T$, we iteratively move in the direction of $\nabla_{x_t} \log p(x_t|y)$. We can derive an equivalent form:

$$p(x_t|y) = \frac{p(y|x_t)p(x_t)}{Z} \tag{29}$$

$$\log p(x_t|y) = \log p(x_t) + \log p(y|x_t) - \log Z \tag{30}$$

$$\nabla_{x_t} \log p(x_t|y) = \nabla_{x_t} \log p(x_t) + \nabla_{x_t} \log p(y|x_t) - \nabla_{x_t} \log Z \tag{31}$$

$$s_\lambda(x_t|y, t) = s_\theta(x_t, t) + \nabla_{x_t} \log p(y|x_t). \tag{32}$$

## C CoDe with Image-Conditioning: CoDe($\eta$)

For (T+I)2I cases, where the reward depends on a target image, the reward distribution deviates significantly from the base distribution $p$. Here, sampling-based approaches would require a relatively larger value of $N$ to achieve alignment. To tackle this, a reference target image $x_{\text{ref}}$, such as a specific style or even stroke painting, is provided as an additional conditioning input. Inspired by image editing techniques using diffusion (Meng et al., 2021; Kooh-payegani et al., 2023), we add partial noise corresponding to only $\tau = \eta \times T$ (with $\eta \in (0,1]$) steps of the forward diffusion process, instead of the full noise corresponding to $T$ steps. This is illustrated in line 2 and 3 of Algorithm 2. Then, starting from this noisy version of the reference image $x_\tau$, CoDe($\eta$) progressively denoises the sample for only $\tau$ steps to generate the clean, reference-aligned image $x_0$ (lines 5 to 10). Specifically, for each block of $B$ steps,

---

**Algorithm 2:** CoDe($\eta$)

**Require:** $p$, $T$, $N$, $B$, $x_{\text{ref}}$, $c$, $\eta$
1   Sample conditional initial noise:
2     $\tau = \eta \times T$
3     $x_\tau = \sqrt{\bar{\alpha}_\tau} x_{\text{ref}} + \sqrt{1 - \bar{\alpha}_\tau} z$, $z \sim \mathcal{N}(0, I)$
4   Initialize counter: $s = 1$
5   **for** $t \in [\tau - 1, \cdots, 0]$ **do**
6     **if** $\text{mod}(s, B) = 0$ **then**
7       Sample $N$ times over $B$ steps:
        $\{x_{t-1}^{(n)}\}_{n=1}^N \overset{i.i.d.}{\sim} \prod_{i=t}^{t+B} p(x_{i-1}|x_i)$
8       Compute values of all N samples:
        $\{x_{t-1}^{(n)}\}_{n=1}^N = \{r(\mathbb{E}[x_0|x_{t-1}^{(n)}])\}_{n=1}^N$
9       Select the sample with maximum value:
        $x_{t-1} \leftarrow \underset{\{x_{t-1}^{(n)}\}_{n=1}^N}{\text{argmax}} V(x_{t-1}^{(n)}; p, c)$
10    **end**
11    $s \leftarrow s + 1$
12   **end**
**Return:** $x_0$

---

we unroll the diffusion model $N$ times independently (Algorithm 2, line 7). Then, based on the value function estimation (line 8), select the best sample (line 9) to continue the reverse process until we obtain a clean image at $t = 0$. A key advantage of CoDe($\eta$) is its ability to achieve similar alignment-divergence trade-offs while using a significantly lower value of $N$, as is demonstrated in Section 5. Note that the inner loop of CoDe($\eta$) (lines 5-10) runs for $\tau$ steps (instead of $T$) due to adjustable noise conditioning discussed in the following. For the sake of brevity, we assume $\tau$ to be divisible by $B$; otherwise, we apply the same steps on a last but smaller block. By conditioning the initial noise sample $x_\tau$ on the reference image $x_{\text{ref}}$, we can generate images $x_0$ that better incorporate the characteristics and semantics of the reference image while adhering to the text prompt $c$. As we demonstrate in our experimentation, threshold $\eta$ provides an *extra knob* built in CoDe($\eta$) allowing the user to efficiently trade off divergence for reward. Note that the reward-conditioning of the generated image is inversely proportional to the value of $\eta$. Setting $\eta = 1$ results in $\tau = T$ and fully deactivates the noise conditioning. A byproduct of this conditioning is compute efficiency, as is discussed in Section 8

Given Theorem 2.1 and its proof in Appendix A, we aim to sample from the reward-tilted posterior $\pi_\lambda^*(x_{t-1}|x_t, c)$ in order to optimize the KL-regularized reward maximization objective Eq. (9). In order to perform CoDe's blockwise guidance, we:

1. sample from the prior $p(x_{t-1}|x_t, c)$ using the denoising diffusion process, across all $N$ streams,

2. and then compute the values of each of the $N$ samples using $V(x_{t-1}; p)$

By doing so, the probability of the selected sample $x_{t-1}$ with the highest value (in Alg. 1 Line 8, 2, Line 9) implicitly incorporates the prior distribution $p(x_{t-1}|x_t, c)$ as a Monte-Carlo estimation technique. Additionally, selecting the highest value sample

$$x_{t-1} \leftarrow \underset{\{x_{t-1}^{(n)}\}_{n=1}^N}{\text{argmax}} V(x_{t-1}^{(n)}; p, c) \tag{33}$$

is mathematically equivalent to sampling from the categorical distribution

$$x_{t-1} \overset{i.i.d.}{\sim} Categorical(\{\texttt{softmax}[V(x_{t-1}^n)/\tau]\}), \forall n \in [1, N], \tag{34}$$

where the temperature $\tau \to 0$.

# D   Additional Results for Case Study I

For the sake of completeness, we also study a variant of the GMM setting as discussed in Section 5, where the mean of the reward distribution is equal to the mean of one of the components in the prior distribution, as shown in Fig. 11. The prior distribution $p(\boldsymbol{x})$ is modelled as a 2-dimensional Gaussian mixture model (GMM) $p(\boldsymbol{x}_0) = \sum_{i=1}^{3} w_i \mathcal{N}(\boldsymbol{\mu}_i, \boldsymbol{\sigma}^2 \boldsymbol{I}_2)$, with $\sigma = 2$, $[\boldsymbol{\mu}_1, \boldsymbol{\mu}_2, \boldsymbol{\mu}_3] = [(5,3), (3,7), (7,7)]$, and $\boldsymbol{I}_d$ is an $d$-dimensional identity matrix, as shown in Fig. 11. All mixture components are equally weighted with, i.e., $w_1 = w_2 = w_3 = 0.33$. In contrast to the previous setup, we define the reward distribution as $p(r|\boldsymbol{x}) = \mathcal{N}(\boldsymbol{\mu}_r, \boldsymbol{\sigma}_r^2 \boldsymbol{I}_2)$ with $\boldsymbol{\mu}_r = [5,3]$ and $\boldsymbol{\sigma}_r = 2$. Based on this setup, we train a diffusion model $p_\theta(x)$ to estimate the prior distribution $p(\boldsymbol{x})$. For this we use a 3-layer MLP that takes as input $(\boldsymbol{x}_t, t)$ and predicts the noise $\boldsymbol{\varepsilon}_t$. It is trained over 200 epochs with $T = 1000$ denoising steps. Then, we implement `CoDe`to guide the trained diffusion model to generate samples with high likelihood under the reward distribution.

In Fig. 11, we present the trade-off curves for normalized expected reward (or win rate) versus KL divergence by adjusting the hyperparameters of the respective methods. For the guidance-based methods DPS and UG, the guidance scale is varied between 1 and 50, whereas for the sampling-based methods BoN, SVDD, and `CoDe`, the number of samples $N$ is varied between 2 and 500. Similar to the results in Section 5, we observe `CoDe` achieve the most favorable trade-off between normalized expected reward and KL divergence, with BoN performing closely behind. In the case of win rate vs. KL divergence, BoN demonstrates the best trade-off, consistent with findings from the literature on Language Model (LM) alignment (Mroueh, 2024; Beirami et al., 2024; Gui et al., 2024). Furthermore, guidance-based methods tend to exhibit higher KL divergence, as they often collapse to the mode of the reward distribution when the guidance scale is increased, leading to a reduction in diversity among the sampled data points. For both performance metrics, SVDD-PM achieves a high expected reward or win rate but at the expense of significantly increased divergence, even for smaller values of $N$. Whereas `CoDe` offers the widely sought-after flexibility, allowing users to balance the trade-off by adjusting parameters such as $N$ and $B$.

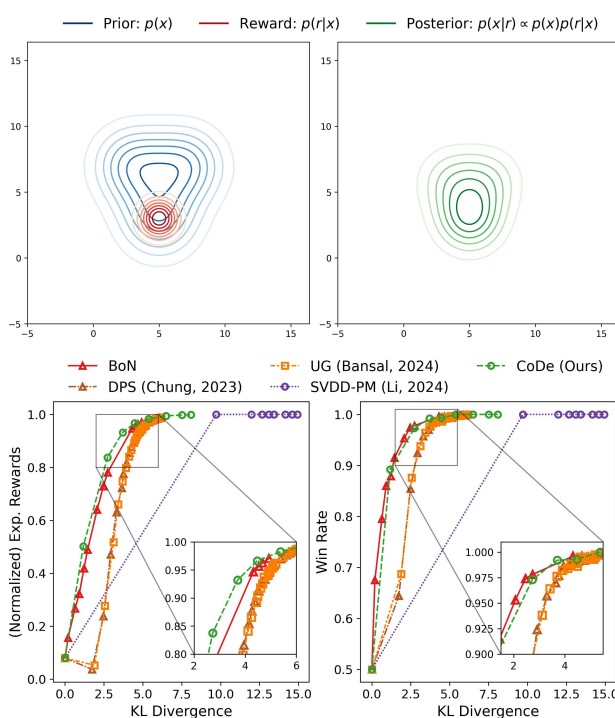

Figure 11: Setup (top row) and reward vs. divergence trade-off (bottom row). `CoDe` offers highest reward at lowest divergence with a much lower $N$ than BoN.

# E    Additional Results for Case Study II

Here, we provide further details about the differentiable reward-guidance scenarios' quantitative evaluations summarized in Table 4 and computational complexity analysis in Table 5.

**Further details on evaluation metrics.** For computing I-Gram, we utilize VGG (Simonyan & Zisserman, 2014) Gram matrices of the reference and generated images to measure image alignment across all scenarios/settings, as commonly followed in the literature (Somepalli et al., 2024; Gatys et al., 2016; Yeh et al., 2020). Specifically, these are computed using the last layer feature maps of an ImageNet-1k pretrained VGG backbone (Simonyan & Zisserman, 2014). Image alignment between a reference, generated image pair is then measured by computing the dot product of their gram matrices. Further, we report a recently proposed CLIP-based Maximum Mean Discrepancy (CMMD) (Jayasumana et al., 2024) as a divergence measure. It overcomes the drawback of FID stemming from the underlying Gaussian assumption in the representation space of the Inception model (Szegedy et al., 2015).

**Qualitative performance.** Let us start with style guidance in Fig. 12. As can be seen, `CoDe` either stands on-par or performs better as compared to all other baselines in terms of capturing both, the style of the reference image and the semantics of the text prompt. This can be seen in comparison with UG for the text prompt of "portrait of a woman", where UG fails to incorporate the text prompt, but latches onto the style of the reference image. The results for face guidance with and without noise-conditioning are illustrated in Figs. 15, 16, respectively. It can be noticed that the noise-conditioned baselines capture the reference face much better than their non noise-conditioned counterparts. Moreover, in the case of noise-conditioning, BoN($\eta$), SVDD-PM($\eta$) and UG($\eta$) fail to meaningfully capture the semantics of the text-prompt, particularly for "Headshot of a woman made of marble". However, `CoDe`($\eta$) captures both, the reference face and the text prompt. In the case of the other text prompt "Headshot of a person with blonde hair with space background", SVDD-PM($\eta$) and `CoDe`($\eta$) offer best results as compared to other baselines. Finally, the results for stroke guidance without noise-conditioning are illustrated in Fig. 17. It can be seen that none of the baselines capture the reference strokes or their color palettes successfully, but only adhere to the text-prompt. This empirically corroborates the need for using noise-conditioning for guidance, when the reward distribution (strokes in this scenario) differs significantly from the base diffusion model's distribution.

**Quantitative performance.** In this section, we break down the quantitative performance of all methods across the three different differentiable reward scenarios of style, face and stroke guidance. We summarize the results in Tab. 6, 7, 8 with the first row corresponding to the base Stable Diffusion model and R: indicating the reward metric used for guiding the diffusion model. For differentiable guidance scenarios, we observe best reward vs divergence/text-alignment tradeoffs using $N, B = 100, 5$ for `CoDe`, $\eta = 0.6$ for `CoDe`($\eta$) in the style and stroke guidance scenarios, $\eta = 0.7$ for face guidance and $\eta = 0.8$ for compression guidance. Similarly for SVDD-PM and BoN, using $N = 100$ renders best results in terms of reward-aligned generated images without over-optimization (low text alignment or high divergence from base distribution as shown in Figs. 19, 20) for differentiable rewards. For compression guidance, we observe best results with $N = 100$ for BoN and $N = 40$ for SVDD-PM, `CoDe` for the T2I scnerario, and $N = 100$ for BoN($\eta$), $N = 7$ for SVDD-PM($\eta$), $N = 40$ for `CoDe`($\eta$) for the (T+I)2I scenario. We observe best results for UG with a guidance scale of 6, 6 forward gradient steps ($m$) and 12 refinement steps ($K$). These have also been reported to work best for style and face guidance by the authors (Bansal et al., 2024b) and we observe these to work best for stroke guidance too. DPS uses a gradient guidance scale of 1.5 for style and stroke guidance and a scale of 200 for face guidance. In the case of MPGD and Freedom used for aesthetic guidance, we observe best results for $\rho = 12.5, 0.15$, respectively, and vary $\rho = [8.5, 10.5, 12.5, 15.5, 17.5]$ for MPGD and $\rho = [0.1, 0.15, 0.2, 0.25, 0.3]$ for Freedom.

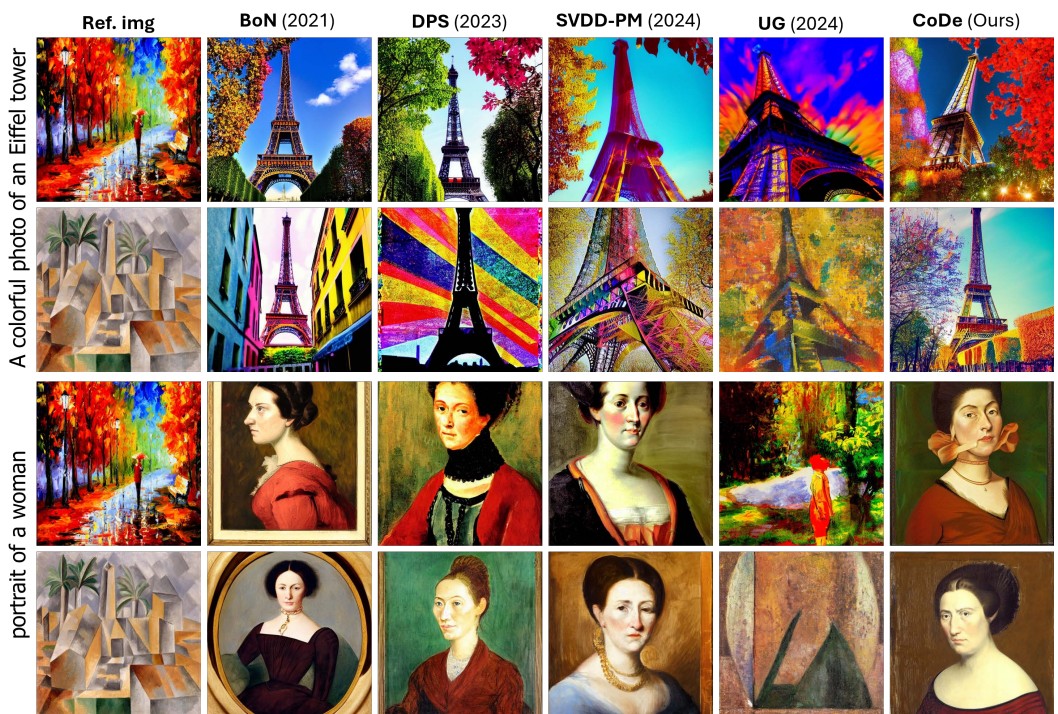

Figure 12: Quality evaluation across methods for style guidance without noise-conditioning.

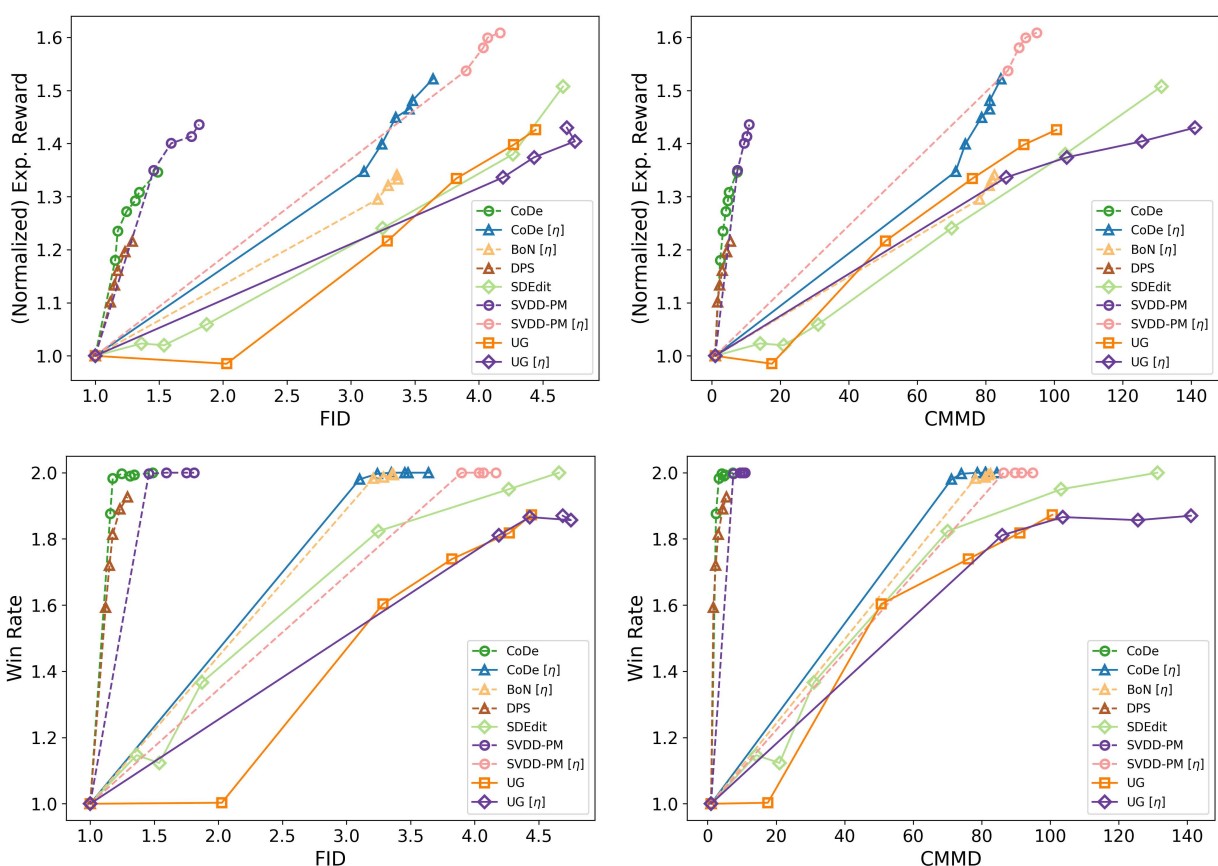

Figure 13: Reward vs. divergence trade-off curves for style guidance.

**Style Guidance**. The results are summarized in Table 6. Compared to the sampling-based guidance counterparts BoN and BoN($\eta$), `CoDe` achieves a higher reward at the cost of slightly higher divergence (FID and CMMD), with and without the noise-conditioning. Yet, with a slightly smaller reward `CoDe`, `CoDe`($\eta$) offers a better performance than SVDD-PM, SVPP-PM($\eta$) across FID, CMMD and T-CLIP. Compared to guidance-based counterparts such as DPS, DPS($\eta$) and UG, UG($\eta$), `CoDe`, `CoDe`($\eta$) offer a better trade-off in terms of reward vs base distribution divergence and reward vs text, image alignment. This is also illustrated in Fig. 13 where `CoDe`($\eta$) consistently outperforms UG, UG($\eta$) in terms of image alignment (normalized expected reward as well as win rate), while offering lesser divergence w.r.t. both FID and CMMD.

Table 6: Quantitative metrics for style guidance.

| Method | R1: Style Guidance | | | | |
|---|---|---|---|---|---|
| | Rew. (↑) | FID (↓) | CMMD (↓) | T-CLIP (↑) | I-Gram (↑) |
| Base-SD (2021) | 1.0 | 1.0 | 1.0 | 1.0 | 1.0 |
| SDEdit (2021) | 1.22 | 3.25 | 67.75 | 0.90 | 1.51 |
| BoN (2022) | 1.14 | 1.30 | 2.25 | 0.99 | 1.1 |
| BoN ($\eta = 0.6$) | 1.34 | 3.36 | 84.02 | 0.87 | 1.57 |
| SVDD-PM (2024) | 1.44 | 1.81 | 10.93 | 0.99 | 1.6 |
| SVDD-PM ($\eta = 0.6$)(2024) | 1.60 | 4.16 | 96.52 | 0.82 | 3.5 |
| DPS (2023) | 1.22 | 1.29 | 5.46 | 0.99 | 1.2 |
| DPS ($\eta = 0.6$)(2023) | 1.29 | 3.31 | 90.06 | 0.83 | 2.5 |
| UG (2024b) | 1.39 | 4.27 | 91.13 | 0.82 | 2.9 |
| UG ($\eta = 0.7$) | 1.37 | 4.43 | 103.6 | 0.79 | 3.5 |
| `CoDe` | 1.34 | 1.49 | 7.40 | 1.0 | 1.6 |
| `CoDe`($\eta = 0.6$) | 1.52 | 3.64 | 84.45 | 0.86 | 3.2 |

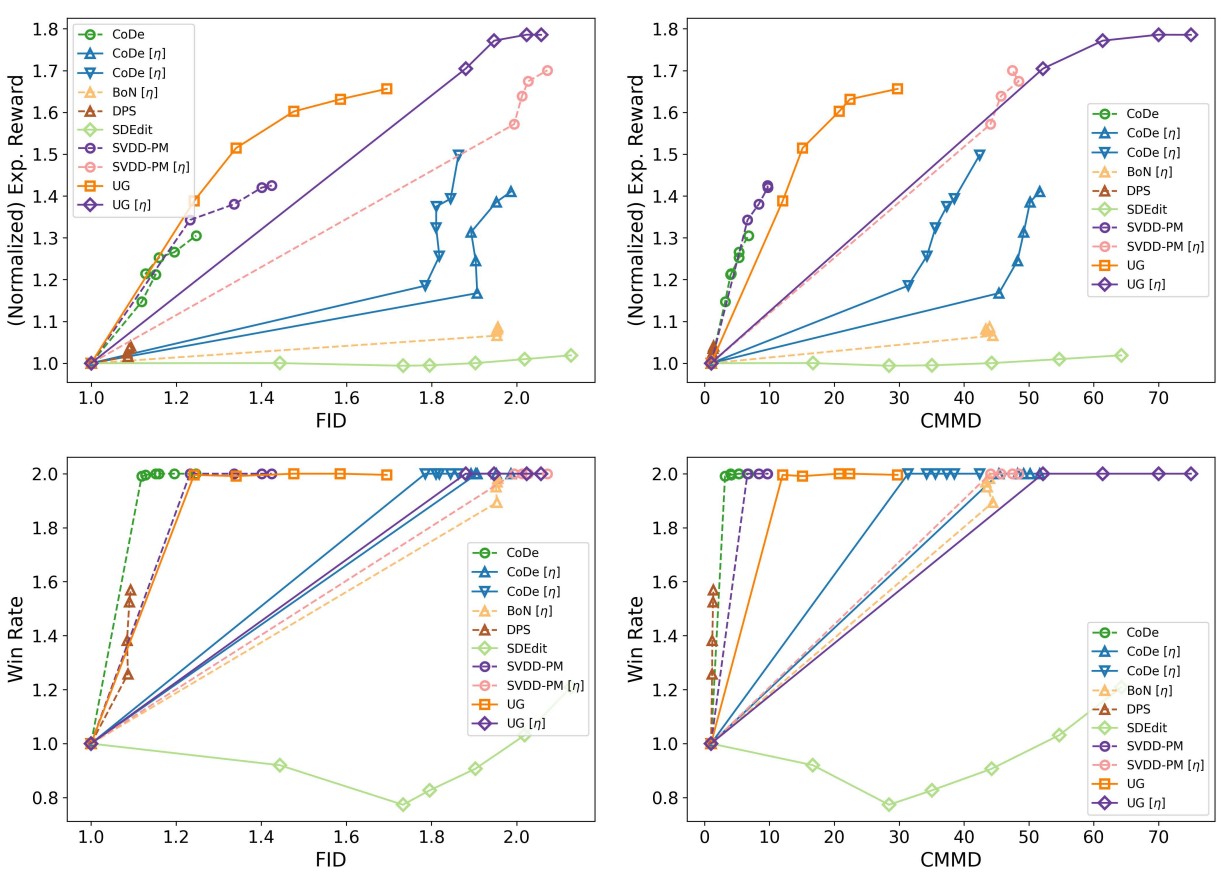

Figure 14: Reward vs. divergence trade-off curves for face guidance.

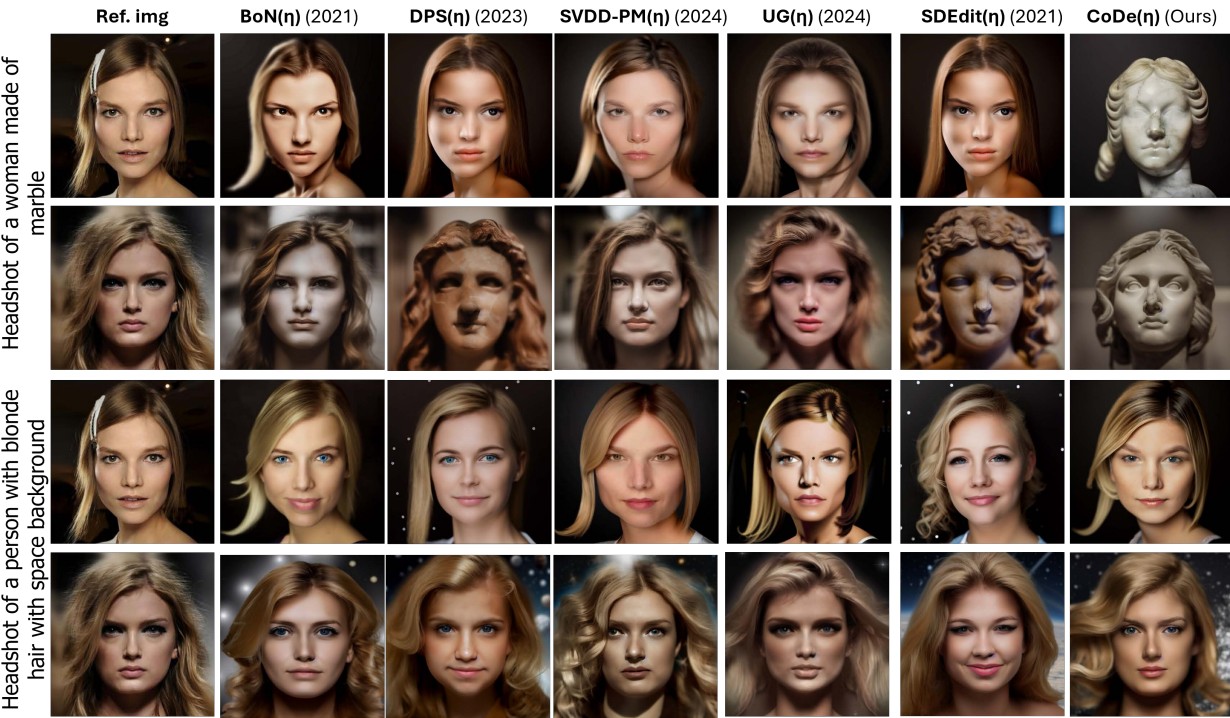

Figure 15: Quality evaluation across methods for face guidance.

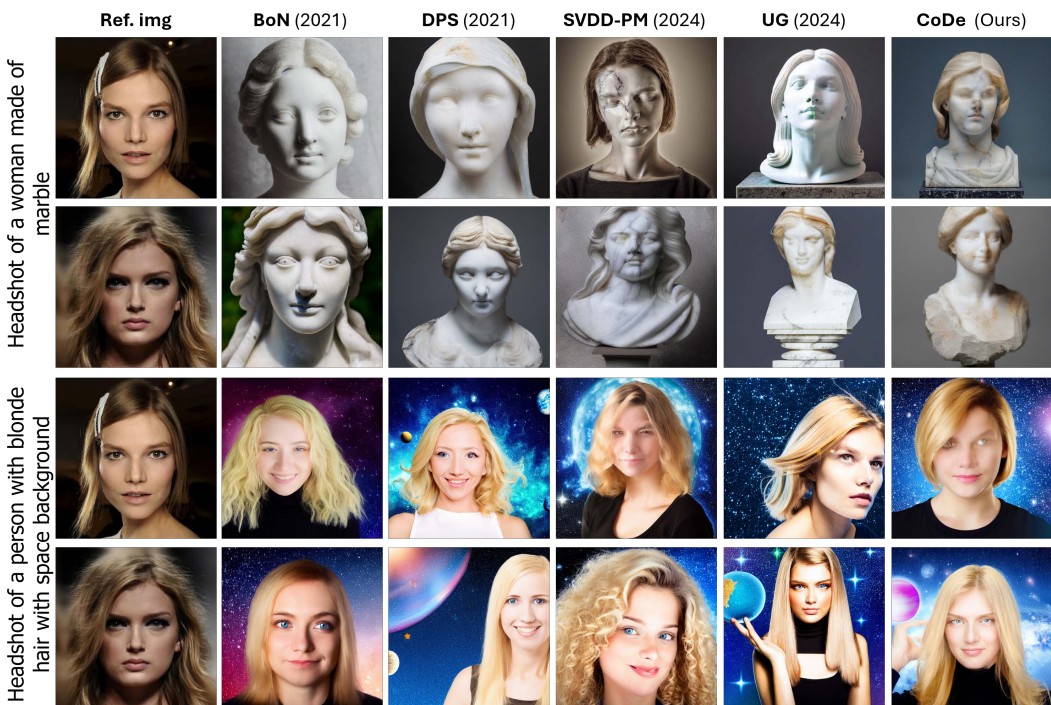

Figure 16: Quality evaluation across methods for face guidance without noise-conditioning.

**Face Guidance**. We summarize the results in Table 7. As the rewards are negative, we first compute the negative log of the reward values and then normalize it with respect to the base.

Table 7: Quantitative metrics for face guidance.

| Method | R2: Face Guidance | | | | |
|---|---|---|---|---|---|
| | **Rew.** (↑) | **FID** (↓) | **CMMD** (↓) | **T-CLIP** (↑) | **I-Gram** (↑) |
| Base-SD (2021) | 1.0 | 1.0 | 1.0 | 1.0 | 1.0 |
| SDEdit (2021) | 0.99 | 1.79 | 34.91 | 0.89 | 1.74 |
| BoN (2022) | 1.08 | 1.22 | 2.52 | 0.99 | 1.0 |
| BoN ($\eta = 0.7$) | 1.08 | 1.82 | 35.3 | 0.88 | 1.8 |
| SVDD-PM (2024) | 1.42 | 1.42 | 9.67 | 0.97 | 0.74 |
| SVDD-PM ($\eta = 0.7$) (2024) | 1.70 | 2.07 | 48.22 | 0.86 | 1.77 |
| DPS (2023) | 1.04 | 1.09 | 1.36 | 0.99 | 1.03 |
| DPS ($\eta = 0.7$) (2023) | 1.21 | 1.71 | 33.21 | 0.86 | 1.68 |
| UG (2024b) | 1.66 | 1.69 | 29.76 | 0.86 | 1.06 |
| UG ($\eta = 0.7$) | 1.77 | 1.94 | 61.27 | 0.85 | 1.78 |
| **CoDe** | 1.30 | 1.25 | 6.76 | 0.98 | 0.91 |
| **CoDe**($\eta = 0.7$) | 1.5 | 1.86 | 42.40 | 0.88 | 1.91 |

Compared to BoN, BoN($\eta$) and DPS, DPS($\eta$), CoDe, CoDe($\eta$) provides higher rewards but also with higher divergence (FID and CMMD). Although SVDD-PM, SVDD-PM($\eta$) and UG, UG($\eta$) achieve higher rewards, CoDe, CoDe($\eta$) offer a better trade-off in terms of FID, CMMD and T-CLIP. Moreover, CoDe($\eta$) offers the best image-alignment in terms of I-Gram as compared to all other baselines.

Additionally, CoDe($\eta$) provides competitive results as compared to UG, which is the second-best method while offering better prompt alignment as reflected in a higher T-CLIP score. We draw similar conclusions from the reward vs. divergence curves presented in Fig. 14, where CoDe($\eta$) achieves competitive rewards as compared to UG, UG($\eta$), SVDD-PM, SVDD-PM($\eta$), but on-par win rates as compared to UG, at the cost of slightly higher FID and CMMD scores.

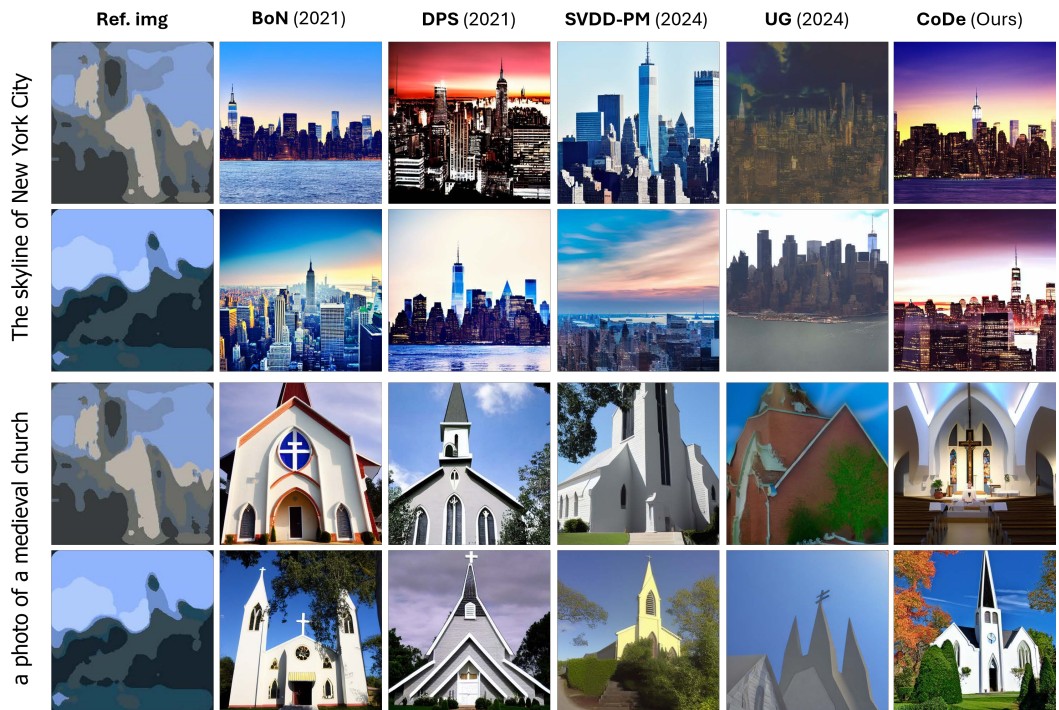

Figure 17: Quality evaluation across methods for stroke guidance without noise-conditioning.

**Stroke**. As shown in Table. 8, among the sampling-based methods, `CoDe` provides better results than BoN in terms of expected reward and FID while maintaining the same T-CLIP score. Although UG and SVDD-PM offer higher rewards, `CoDe` offers lower divergence (FID and CMMD) and better T-CLIP scores. Overall, we observe that `CoDe`($\eta$) has the highest rewards while offering competitive FID, CMMD and T-CLIP.

Table 8: Quantitative metrics for stroke generation.

| Method | R3: Stroke Generation | | | | |
|---|---|---|---|---|---|
| | **Rew.** ($\uparrow$) | **FID** ($\downarrow$) | **CMMD** ($\downarrow$) | **T-CLIP** ($\uparrow$) | **I-Gram** ($\uparrow$) |
| Base-SD (2021) | 1.0 | 1.0 | 1.0 | 1.0 | 1.0 |
| SDEdit (2021) | 1.38 | 2.79 | 145.6 | 0.90 | 2.64 |
| BoN (2022) | 1.25 | 1.05 | 4.5 | 0.99 | 1.12 |
| BoN ($\eta = 0.6$) | 1.55 | 3.12 | 170 | 0.89 | 3.05 |
| SVDD-PM (2024) | 1.56 | 1.04 | 12.0 | 0.99 | 1.38 |
| SVDD-PM ($\eta = 0.6$) (2024) | 1.83 | 3.87 | 187.1 | 0.85 | 4.4 |
| DPS (2023) | 1.34 | 1.04 | 14.0 | 0.97 | 1.13 |
| DPS ($\eta = 0.6$) (2023) | 1.45 | 2.81 | 195.0 | 0.88 | 2.83 |
| UG (2024b) | 1.55 | 2.78 | 78.0 | 0.88 | 1.63 |
| UG ($\eta = 0.6$) | 1.66 | 4.45 | 236.5 | 0.78 | 1.21 |
| **CoDe** | 1.41 | 0.78 | 6.5 | 0.99 | 1.38 |
| **CoDe($\eta = 0.6$)** | 1.75 | 3.50 | 178.5 | 0.87 | 4.25 |

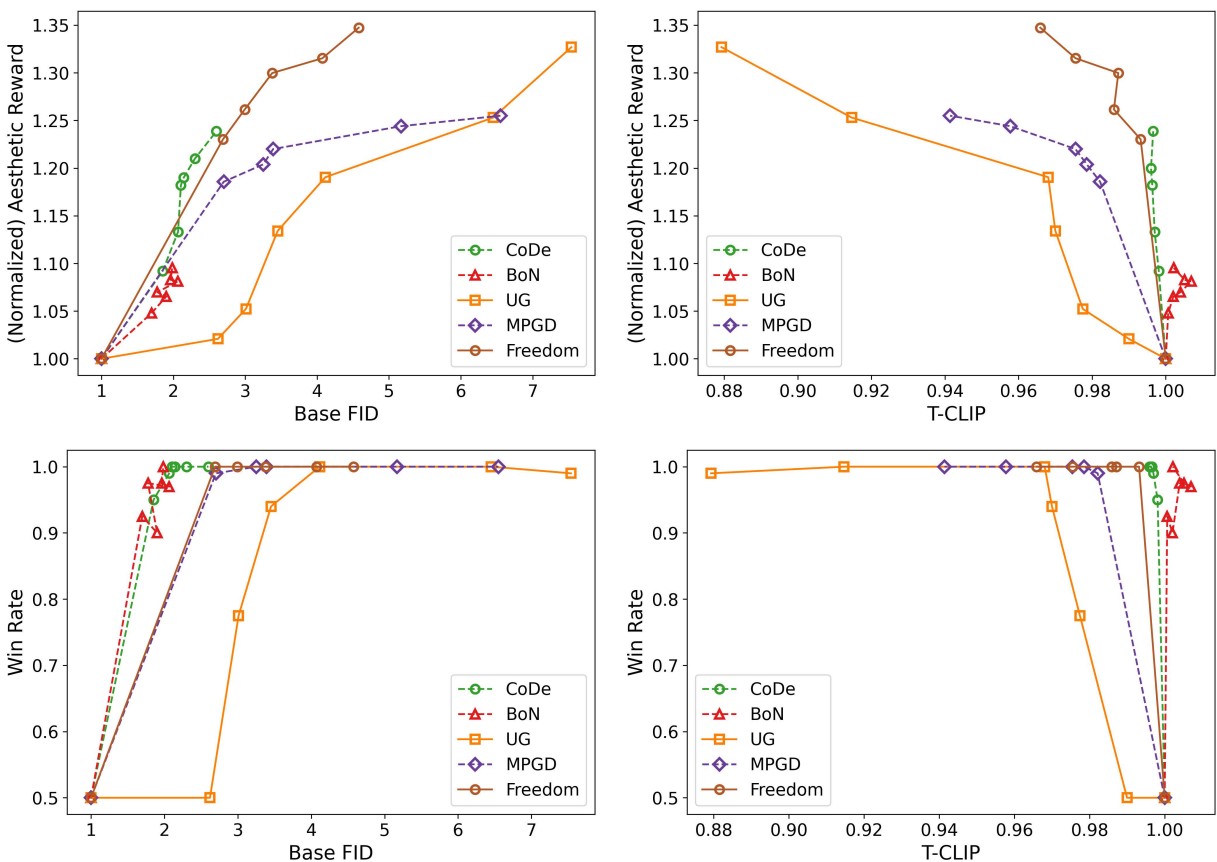

Figure 18: Reward vs. divergence trade-off curves for aesthetic guidance.

**Aesthetic Guidance**. Given the tradeoff curves in Figs. 18, we observe that `CoDe` offers better or on par rewards as compared to MPGD (He et al., 2024) for smaller FID and higher T-CLIP scores, thus offering a better reward vs divergence tradeoff. When compared to Freedom Yu et al. (2023) and UG Bansal et al. (2024b), `CoDe` achieves competitive or lesser rewards but offers better text alignment (T-CLIP) and lower divergence from the base distribution (FID, CMMD). This also corroborates in Fig. 8 where UG generates aesthetic images that do not completely adhere to the text-prompt leading to reward over-optimization.

**Computation Complexity.** We present a breakdown of the computational complexities of all baselines across each of the guidance scenarios. DPS is considerably faster across all three generation scenarios among the gradient-based guidance methods. This is due to the $m$ gradient and $K$ refinement steps used in UG, which are not used in DPS. The difference is more pronounced in the case of style- and stroke guidance as UG uses a higher number of gradient steps $m$. Further, among the sampling-based approaches, SVDD-PM is an order of magnitude slower than BoN as it applies token-wise guidance. On the contrary, our blockwise approaches CoDe, CoDe($\eta$) are more efficient than UG and SVDD-PM and closely follow BoN.

Table 9: Computational Complexity

| Methods | Inf. Steps | Rew. Queries | Runtime [sec/img] | | |
|---|---|---|---|---|---|
| | | | Style | Face | Stroke |
| Base-SD 2021 | $T$ | - | 14.12 | 14.12 | 14.12 |
| BoN 2022 | $NT$ | $N$ | 266.02 | 268.43 | 265.86 |
| SVDD-PM 2024 | $NT$ | $NT$ | 1168.74 | 1859.67 | 1169.68 |
| DPS 2023 | $T$ | $T$ | 62.52 | 122.21 | 61.83 |
| UG 2024b | $mKT$ | $mKT$ | 1588.41 | 543.12 | 1592.89 |
| CoDe | $NT$ | $NT/B$ | 441.81 | 583.12 | 442.08 |
| CoDe($\eta$) | $N\eta T$ | $N\eta T/B$ | 331.42 | 403.19 | 274.56 |

# F  Details for Estimating KL Divergence

To compute the KL divergence between the guided and the base diffusion model, we draw on some existing results that give us an upper bound on the KL divergence between `CoDe` and the base diffusion model, which is given by the following:

**Lemma F.1.**

$$KL(\texttt{CoDe}(N, B) \,\|\, Base) \leq \left( \log N - \frac{N-1}{N} \right) \times \frac{T}{B}. \tag{35}$$

*Proof.* The proof follows the same lines as (Beirami et al., 2024, Theorem B.1), with the exception that we need to resort to (Mroueh, 2024, Theorem 1) to bound the KL divergence of each intervention. □

For BoN where the block size $B = T$, the KL divergence is upper bounded by

$$KL(\text{BoN}(N) \,\|\, \text{Base}) \leq \log N - \frac{N-1}{N},$$

which is directly implied by (Mroueh, 2024, Theorem 1) as well. For SVDD-PM where $B = 1$, the KL divergence is upper bounded by

$$KL(\text{SVDD-PM}(N) \,\|\, \text{Base}) \leq \left( \log N - \frac{N-1}{N} \right) \times T.$$

Since the noise-conditioned variants of these methods only denoise for $\eta T$ steps instead of the full $T$ steps, the KL divergences are upper bounded using

$$KL(\texttt{CoDe}(N, B, \eta) \,\|\, \text{Base}) \leq \left( \log N - \frac{N-1}{N} \right) \times \frac{\eta T}{B}, \tag{36}$$

$$KL(\text{BoN}(N, \eta) \,\|\, \text{Base}) \leq \log N - \frac{N-1}{N}, \tag{37}$$

$$KL(\text{SVDD-PM}(N, \eta) \,\|\, \text{Base}) \leq \left( \log N - \frac{N-1}{N} \right) \times \eta T. \tag{38}$$

## F.1  Numerical computation of KL divergence for Gaussian models (Case Study I)

In Section 5, to estimate the KL divergence between the base and guided models, we first generate 1000 samples from the base diffusion model and the reward guided model each. Then assuming Gaussian densities for both, we compute the mean and variance for each of the distributions and then use the closed-form expression to calculate the KL divergence between two Gaussians. We notice that in this setting when we reach the degeneracy limit, the bounds suggested by Lemma F.1 are loose, particularly for all SVDD-PM experiments in Section 5. This is a known issue with these KL bounds and has been discussed by Beirami et al. (2024).

## G    General Guidelines for Setting `CoDe`'s $N, B, \eta$

`CoDe` utilizes three parameters $N, B, \eta$ in order to guide the diffusion denoising process towards a reward-tilted posterior. The interplay between these three parameters has been demonstrated through various reward vs divergence tradeoff curves, reward vs text alignment tradeoff curves (Figs. 13, 14, 18) and performance vs efficiency tradeoff curves (Fig. 10). In this section, we discuss the impact of each parameter on the guidance process and then provide a general set of guidelines on how to choose these values based on different tasks.

- **$N, B$**:

    Intuitively, $N$ and $B$ impact the exploration of the prior distribution thus controlling the chances of sampling from a higher reward region (modes of the reward-distribution) while denoising.

    Practically, increasing $N$ leads to higher rewards or more reward-aligned generated images. However, increasing $N$ also leads to a higher divergence from the base distribution (FID, CMMD, KL Divergence), lower text-alignment (T-CLIP) and a linear increase in computational complexity (inference steps and reward queries).

    On the other hand, reducing $B$ increases the number of times the denoising process is diverted towards a high reward region in its distribution thus increasing reward-alignment in generated images. Reducing $B$ also leads to a higher divergence from the base distribution (FID, CMMD, KL Divergence), lower text-alignment (T-CLIP) and a linear increase in computational complexity (inference steps and reward queries).

    The divergence increases logarithmically in $N$ (Eq. (36)) and the compute increases linearly in $N$ (Tab. 9).

    On the other hand, divergence and compute both increase exponentially as $B$ reduces (Eq. (36), Tab. 9).

- **$\eta$** :

    Intuitively, $\eta$ controls the degree of conditioning of the input reference image on the generated image. For a smaller $\eta$, the denoising process starts from a slightly noised version of the input reference image and only denoises the image for $\eta T$ steps instead of the full $T$ steps, thus also reducing the total number of steps that could lead to reward-alignment (Section C Alg. 1, 2).

    Thus, in practice, reducing $\eta$ leads to an increase in reference image alignment and a reduction in reward and text alignment. In cases where the reference image is sampled from the reward distribution (style, face and stroke guidance in (T+I)2I settings with `CoDe($\eta$)`), reducing $\eta$ leads to an increase in reward alignment.

    The computational complexity varies linearly with $\eta$.

Depending on the nature of the task and the divergence of the reward distribution from the prior, the guidelines mentioned above can be used to increase/decrease $N$, $B$, $\eta$ for the desired tradeoffs.

# H  Adaptive Control on $N, B$ for CoDe

Various related works (Raya & Ambrogioni, 2023; Kim et al., 2023; Sclocchi et al., 2025) have empirically demonstrated that different denoising steps have varying impacts on the final output. The early stages primarily determine the overall structure of the image, whereas the latter stages refine the fine-grained details. Given this observation, CoDe could potentially improve its performance by incorporating an adaptive control mechanism that adjusts its blockwise sampling strategy based on the denoising stage. In order to asses whether this adaptive strategy could help CoDe further improve its performance, we perform an ablation experiment by varying $N, B$ across the first half $T_1 \in [500, 250]$ and second half $T_2 \in (250, 0]$ of the diffusion denoising process, for the T2I compression guidance scenario. We experiment on a total of four cases by varying $N, B$ across the two halves of denoising. In the first two cases, $N = 40$ throughout denoising and $B = 1$ for $T_1$, $B = 5$ for $T_2$ or $B = 5$ for $T_1$, $B = 1$ for $T_2$ (rows with $N = 40, B_1 = 1, B_2 = 5$ and $N = 40, B_1 = 5, B_2 = 1$ in Tab. 10). In the next two cases, $B = 5$ throughout denoising and $N = 40$ for $T_1$, $N = 20$ for $T_2$ or $N = 20$ for $T_1$, $N = 40$ for $T_2$ (rows with $N_1 = 40, N_2 = 20, B = 5$ and $N_1 = 20, N_2 = 40, B = 5$ in Tab. 10). We observe that hyperparameters that enforce stronger reward-guidance in the second half of denoising ($T_2$) offer higher rewards as compared to the ones which enforce it in the first half ($T_1$). $N_1 = 40, N_2 = 20, B = 5$ offers lower rewards, lower divergence (FID, CMMD) and on-par text alignment (T-CLIP) as compared to $N = [20, 40], B = 5$. Similar observations extend for ablating on $B$ where $N = 40, B_1 = 1, B_2 = 5$ offers lower reward, lower divergence (FID, CMMD) and slightly lower text alignment (T-CLIP) as compared to $N = 40, B_1 = 5, B_2 = 1$. We would like to clarify that except for the experiments where we swap $N_1$ and $N_2$ or $B_1$ and $B_2$, the ratio of $N/B$ does not remain the same. Note that using a higher $N$ or lower $B$ throughout denoising to exert stronger inference-time guidance results in higher rewards and higher divergence as compared to their adaptive control counterparts. These observations for using stronger guidance in the second half of denoising intuitively align with the task of generating compressible images, where the fine-grained details and the texture of the image is to be controlled. But for other rewards that demand control over the structure and positioning of objects in generated images, a stronger reward-guidance signal in the first half of denoising might be more effective.

Table 10: Quantitative metrics for compression reward with adaptive control.

| Method | Compressibility Reward - T2I | | | | |
|---|---|---|---|---|---|
| | H.P. | Rew. (↑) | FID (↓) | CMMD (↓) | T-CLIP (↑) |
| Base-SD | - | 1.0 | 1.0 | 1.0 | 1.0 |
| BoN | $N = 100, B = T$ | 1.23 | 1.10 | 1.70 | 0.99 |
| SVDD-PM | $N = 40, B = 1$ | 1.83 | 2.86 | 61.75 | 0.88 |
| CoDe | $N = 20, B = 5$ | 1.54 | 2.01 | 12.83 | 0.97 |
| CoDe | $N = 40, B = 5$ | 1.65 | 2.12 | 32.70 | 0.95 |
| CoDe | $N = 40, B_1 = 1, B_2 = 5$ | 1.69 | 2.36 | 42.51 | 0.94 |
| CoDe | $N = 40, B_1 = 5, B_2 = 1$ | 1.76 | 2.41 | 48.30 | 0.95 |
| CoDe | $N_1 = 40, N_2 = 20, B = 5$ | 1.58 | 2.08 | 15.35 | 0.97 |
| CoDe | $N_1 = 20, N_2 = 40, B = 5$ | 1.60 | 2.11 | 17.97 | 0.97 |

# I  Reward Over-Optimization in Compression Guidance for SVDD-PM($\eta$)

Following section 6.1, Fig. 5, we demonstrate a few images generated in the compressibility guidance scenario with SVDD-PM($\eta$) for $N = [20, 30, 40, 100]$, where reward over-optimization occurs. As can be seen in Fig. 19, higher values of $N$ for SVDD-PM($\eta$)'s guidance lead to degenerate generation of images, where the text prompt and reference image alignment is compromised at the cost of high compressibility reward. The generated images roughly follow the color palette of the reference image but fail to meaningfully incorporate the style and aesthetics of the reference image. Moreover, the images also do not resemble natural images, empirically corroborating the high KL-divergence w.r.t. the base distribution in Fig. 5.

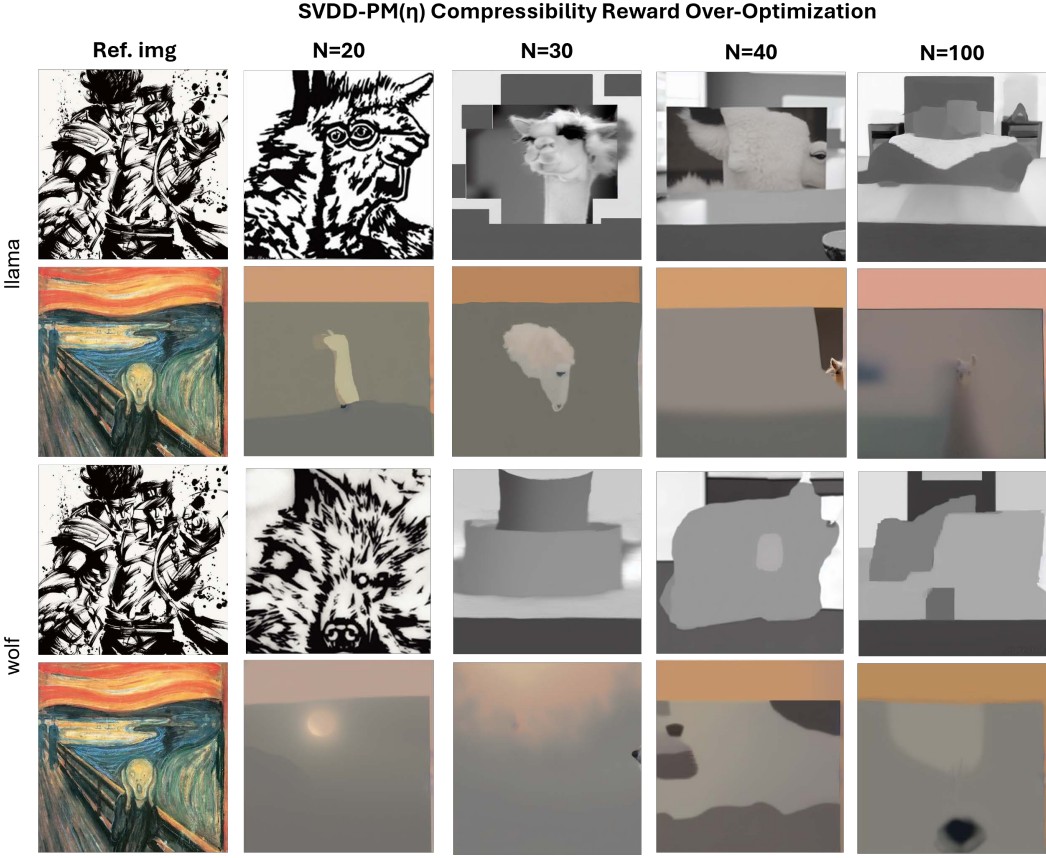

Figure 19: Qualitative examples of reward over-optimized images from SVDD-PM($\eta$) for $N = [20, 30, 40, 100]$, in the compression guidance scenario.

## J    UG with a high guidance scale offers low text alignment

Following section 6.3, Fig. 9, we illustrate a few generated samples of UG across four settings for style guidance with higher guidance scales of 12 and 24 to qualitatively corroborate their low text-alignment. As can be seen in Fig. 20, the generated images offer high alignment with respect to the reference image but fail to incorporate any meaningful features of the text prompts. None of the generated images resemble the Eiffel tower or the portrait of a woman.

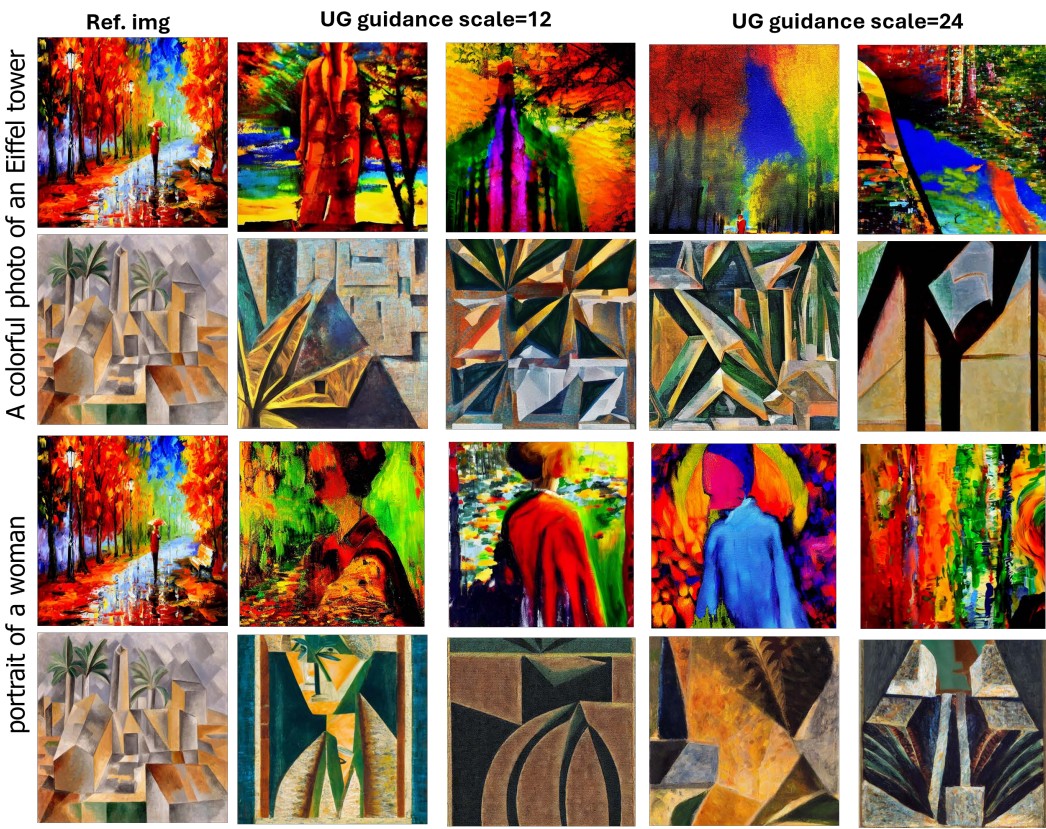

Figure 20: Qualitative examples of low text-alignment (T-CLIP) for UG with higher guidance scales, in the style guidance scenario.

# K Correlation of I-GRAM with I-CLIP Reward

The reason why we also use I-Gram to indicate reward-alignment (instead of only expected reward per scenario) in our evaluations is because the expected reward has already been *seen* by the model throughout the guidance process. Thus, we test all guidance methods on the *unseen* I-GRAM reward-alignment metric to provide a holistic evaluation. That being said, we do notice slight discrepancies in the behavior of the I-GRAM scores and the CLIP-image similarity reward due to the differences in what each of these metrics capture. The I-GRAM score has been shown to capture similarities in style and texture of two images (Gatys et al., 2016) whereas the CLIP-image similarity score measures semantic similarity between any two images. To analyze the correlation between these two metric qualitatively, we present a few generated images for the style guidance scenario in Fig. 21. As can be seen, for the first and second generated images, the reward and I-GRAM for the second image is higher than the first, indicating alignment between the two metrics. However, for the first and third image, the reward is higher for the third image but vice-versa for I-GRAM. Similar alignment and misalignment patterns can be observed for pairs of other images. We attribute this discrepancy to the texture and semantic differences of the generated and reference images. Additionally, to confirm that I-GRAM can be used as an evaluation metric for reward guidance using CLIP similarity, we computed the Pearson Correlation coefficient between the two metrics across a subset of all generated images in the style guidance scenario. We observe a correlation coefficient of 0.87 between Reward and I-GRAM, indicating a positive correlation / direct proportionality between the two.

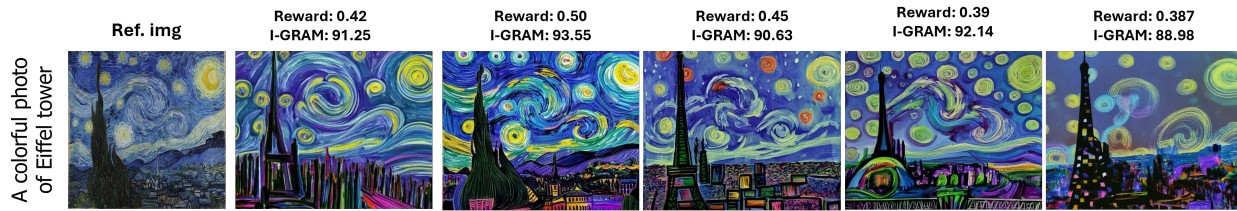

Figure 21: Qualitative demonstration of Reward vs I-GRAM for different style guidance generated images.

## L    Miscellaneous Results

In this section, we illustrate several additional generated images across all baselines and guidance scenarios. We also provide additional results for CoDe, CoDe($\eta$) across various different reference images and text prompt pairs, that are different from the ones already explored in the main manuscript, in Figs. 22, 23, 24.

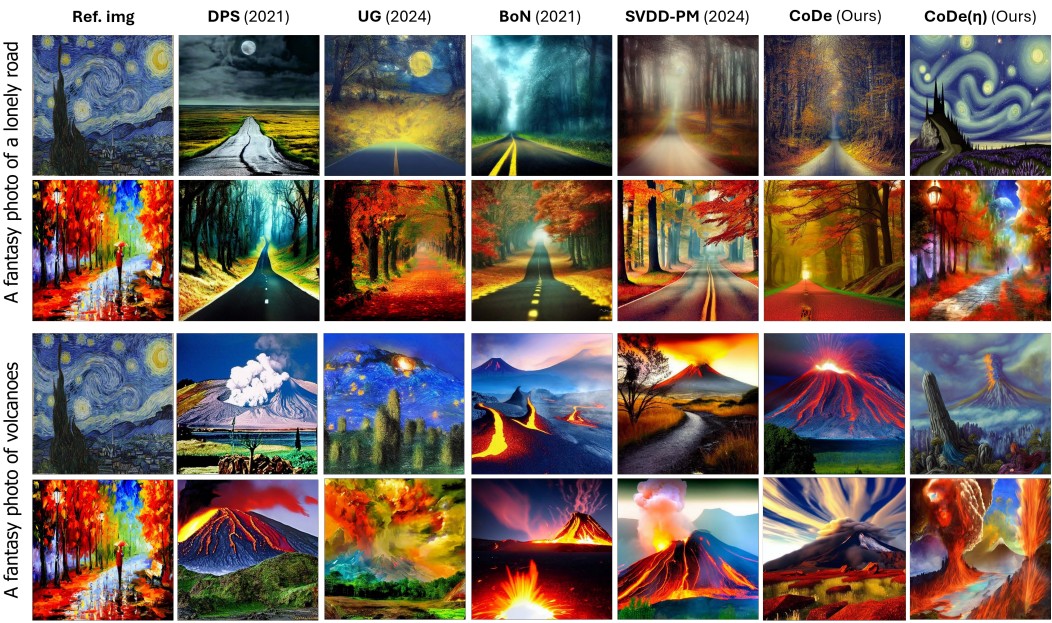

Figure 22: Quality evaluation across methods for style guidance on additional settings without noise-conditioning.

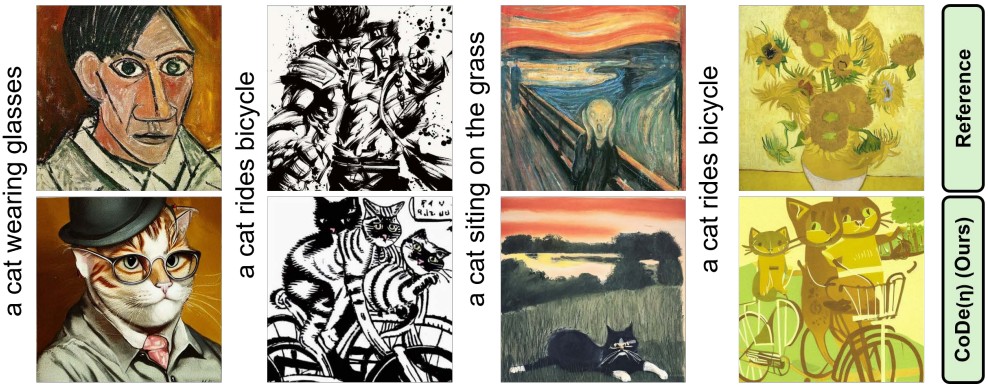

Figure 23: Quality evaluation of CoDe for style guidance on additional settings.

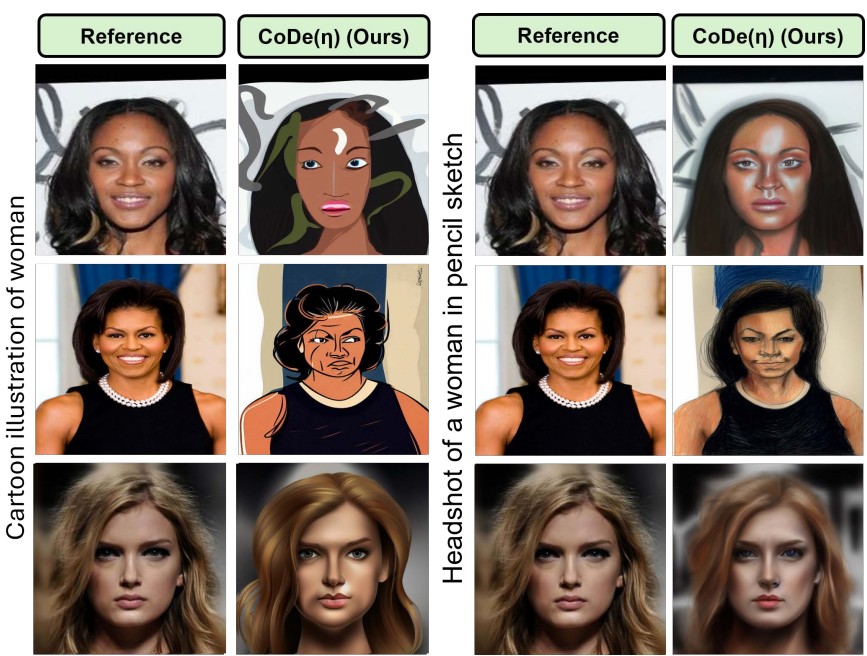

Figure 24: Quality evaluation of `CoDe` for face guidance on additional settings.

