# OpenReview forum: "CoDe: Blockwise Control for Denoising Diffusion Models"
_TMLR — Accepted by TMLR_

### Review · Reviewer_qWvG · 2025-02-26

**Summary Of Contributions:**

This work presents CoDe, a simple yet effective blockwise control method for diffusion models that optimizes reward alignment during inference while avoiding computationally expensive gradient steps. It achieves comparable or better performance than Best-of-N sampling while being much more efficient.

**Audience:**

Yes

**Broader Impact Concerns:**

The author present the broader impact in the paper.

**Claims And Evidence:**

Yes

**Requested Changes:**

1. It is better to add a citation or refrence for the claim in Theorem 2.1.
2. The statement under Eq.(12) is not proper. In Universal Guidance and DPS, they also can use pretrained detection models to extract related features and also use Tweedie's formula to obtain $\hat{x}_0$. In some sense, it can also been seem as a kind of reward function. Claim that they need to use training a separate model undermines their contributions.
3. The samples drawn from $\prod_{i=t}^{t+B} p(x_{i-1}|x_i)$ should be $x_{t+B-1}$ not $x_{t-1}$.
4. For the comparison, there are also more advance gradient based methods, such as FreeDOM [1], SAG [2], Flowgrad [3], where some are based on SD 1.5 and have better performance than UG and DPS. A comparison with these stronger baselines would provide a clearer picture of CoDe’s advantages and limitations.
5. Provide a structured recipe for selecting $N$ and $B$.

[1] FreeDoM: Training-Free Energy-Guided Conditional Diffusion Model
[2] Towards Accurate Guided Diffusion Sampling through Symplectic Adjoint Method.
[3] Flowgrad: Controlling the output of generative odes with gradients

**Strengths And Weaknesses:**

### Strengths:
1. The paper presents Controlled Denoising (CoDe), an inference-time blockwise sampling method to guide diffusion model outputs without requiring gradient computations. This provides a lightweight alternative to finetuning and gradient-based guidance.
2. CoDe achieves comparable reward alignment to best-of-N (BoN) sampling while requiring significantly fewer samples, improving efficiency over traditional BoN methods.
3. Unlike gradient-based methods, CoDe supports both differentiable and non-differentiable rewards, making it more versatile for real-world tasks like image compressibility and stylistic adaptation.

### Weaknesses:
1. Different denoising steps have varying impacts on the final output. The early stages primarily determine the overall structure of the image, while the later stages refine fine-grained details. CoDe could potentially improve its performance by incorporating adaptive control mechanisms that adjust sampling strategies based on the denoising stage.
2. CoDe is not sufficiently compared with the more advanced gradient-based models for differentiable rewards.
3. The effectiveness of CoDe depends on key hyperparameters such as $N$ and $B$, but the paper does not provide a clear guideline on how to choose these values based on different tasks. Offering a structured recipe or heuristic for selecting $N$ and $B$ could greatly enhance the usability of CoDe.

---

> ### Author Response · Authors · 2025-03-23
> **Response to Reviewer qWvG**
>
> > W1. Different denoising steps have varying impacts on the final output. The early stages primarily determine the overall structure of the image, while the later stages refine fine-grained details. CoDe could potentially improve its performance by incorporating adaptive control mechanisms that adjust sampling strategies based on the denoising stage.
>
> A1: Thanks for your insightful  feedback. Following your suggestion, we are now performing an ablation on CoDe guidance with a varying $N$, $B$ across the first half $T \in [500,250]$ and second half $T \in [250,0]$ of the diffusion denoising process. We will add the results of this ablation in the Appendix of the revised manuscript in about a week or two upon completion and analysis of the experiments.
>
> > W2. CoDe is not sufficiently compared with the more advanced gradient-based models for differentiable rewards.
>
> A2: Thanks for your constructive feedback. Following your suggestion, we now also compare CoDe against MPGD [1] and FreeDoM[2] (two recent, advanced gradient-based guidance methods) on the aesthetic guidance scenario. The results for this have now been added in the Section 6.2, Table 3, Fig.8 and the reward vs divergence, reward vs text-alignment trade-off curves have been added in the Appendix E, Fig.18 of the revised manuscript. We observe that CoDe offers better rewards as compared to MPGD and BoN while being second to best as compared to Freedom and UG . However, CoDe offers better text alignment (T-CLIP) and a significantly lower divergence from the base distribution (FID, CMMD) as compared to all its gradient based counterparts, thus offering a better overall reward vs divergence tradeoff. This can also be observed in Fig.18, where CoDe offers almost the same reward as Freedom and UG, but at a lower divergence or higher T-CLIP.
>
> > W3. The effectiveness of CoDe depends on key hyperparameters such as N and B, but the paper does not provide a clear guideline on how to choose these values based on different tasks. Offering a structured recipe or heuristic for selecting N and B could greatly enhance the usability of CoDe.
>
> A3: Thanks for the insightful suggestion. Following your remark, we have now added a section in Appendix G, General Guidelines for Setting CoDe’s hyperparameters $N$, $B$, $\eta$, of the revised manuscript. This provides an intuitive explanation about the impact of each parameter on reward guidance, divergence from the prior, text alignment and computational complexity and how to practically use them which we hope will make it easier to tune these hyperparameters by practitioners.
>
> In the following, we discuss the impact of each parameter on the guidance process and then provide a general set of guidelines on how to choose these values based on different tasks.
>
> 1. $N, B$:
>
> Intuitively, $N$ and $B$ impact the exploration of the prior distribution thus controlling the chances of sampling from a higher reward region (modes of the reward-distribution) while denoising.
>
> Practically, increasing $N$ leads to higher rewards or more reward-aligned generated images. However, increasing $N$ also leads to a higher divergence from the base distribution (FID, CMMD, KL Divergence), lower text-alignment (T-CLIP) and a linear increase in computational complexity (inference steps and reward queries).
>
> On the other hand, reducing $B$ increases the number of times the denoising process is diverted towards a high reward region in its distribution thus increasing reward-alignment in generated images. Reducing $B$ also leads to a higher divergence from the base distribution (FID, CMMD, KL Divergence), lower text-alignment (T-CLIP) and a linear increase in computational complexity (inference steps and reward queries).
>
> The divergence increases logarithmically in $N$ (eq.36) and the compute increases linearly in $N$ (Tab.9).
>
>  On the other hand, divergence and compute both increase exponentially as $B$ reduces (eq.36, Tab.9).
>
> 2. $\eta$ :
>
> Intuitively, $\eta$ controls the degree of conditioning of the input reference image on the generated image. For a smaller $\eta$, the denoising process starts from a slightly noised version of the input reference image and only denoises the image for $\eta T$ steps instead of the full $T$ steps, thus also reducing the total number of steps that could lead to reward-alignment (Section C Alg.1,2).
>
> Thus, in practice, reducing $\eta$ leads to an increase in reference image alignment and a reduction in reward and text alignment. In cases where the reference image is sampled from the reward distribution (style, face and stroke guidance in (T+I)2I settings with CoDe($\eta$)), reducing $\eta$ leads to an increase in reward alignment.
>
> The computational complexity varies linearly with $\eta$.

---

> ### Author Response · Authors · 2025-03-23
> **Response to Reviewer qWvG**
>
> > W3.The effectiveness of CoDe depends on key hyperparameters such as N and B, but the paper does not provide a clear guideline on how to choose these values based on different tasks. Offering a structured recipe or heuristic for selecting N and B could greatly enhance the usability of CoDe.
>
> A3. (... Continued here...) Depending on the nature of the task and the divergence of the reward distribution from the prior, the guidelines mentioned above can be used to increase/decrease $N$, $B$, $\eta$ for the desired tradeoffs.
>
>
> **Requested Changes (RC)**
>
> > RC1. It is better to add a citation or reference for the claim in Theorem 2.1.
>
> A1: Theorem 2.1 has been derived in the context of reward-alignment for language models in [3], which we cite in Section 2.3 after Theorem 2.1. Additionally, we also cite a few more relevant works in the RL-based fine-tuning and inference-time guidance literature that utilize or discuss Theorem 2.1. We’d be happy to incorporate more relevant works per your suggestion.
>
> > RC2. The statement under Eq.(12) is not proper. In Universal Guidance and DPS, they also can use pretrained detection models to extract related features and also use Tweedie's formula to obtain $\hat{x}_0$. In some sense, it can also been seen as a kind of reward function. Claim that they need to use training a separate model undermines their contributions.
>
> A2: Thanks for pointing this out. We have fixed this in the revised draft.
>
> > RC3. Confusion with indexing samples in diffusion denoising. (Alg1, Line5)
>
> A3. A: We disagree. Since we index the timesteps during diffusion denoising from T to 0 (in descending order), and not 0 to T, the sample drawn after a block of B denoising steps is from $x_{t-1}$ and not $x_{t+B-1}$. In fact, $x_{t+B-1}$ is the first step of denoising in a block between $i \in [t, t+B]$. We have now added a clarification about this in the explanation of Alg.1 in Section 3.
>
> > RC4. For the comparison, there are also more advanced gradient based methods, such as FreeDOM [1], SAG [2], Flowgrad [3], where some are based on SD 1.5 and have better performance than UG and DPS. A comparison with these stronger baselines would provide a clearer picture of CoDe’s advantages and limitations.
>
> A: Following your suggestion, we now also compare CoDe against MPGD [1] and FreeDoM[2] (two recent, advanced gradient-based guidance methods) on the aesthetic guidance scenario. The results for this have now been added in the Appendix Section E, Table 8 and the reward vs divergence, reward vs text-alignment trade-off curves have been added in Fig.16 of the revised manuscript. We observe that CoDe offers better rewards as compared to MPGD and BoN while being second to best as compared to Freedom and UG . However, CoDe offers better text alignment (T-CLIP) and a significantly lower divergence from the base distribution (FID, CMMD) as compared to all its gradient based counterparts, thus offering a better overall reward vs divergence tradeoff. This can also be observed in Fig.17, where CoDe offers almost the same reward as Freedom and UG, but at a lower divergence or higher T-CLIP.
>
>
> References:
>
> [1] He, Yutong, et al. "Manifold preserving guided diffusion." arXiv preprint arXiv:2311.16424 (2023).
>
> [2] Yu, Jiwen, et al. "Freedom: Training-free energy-guided conditional diffusion model." Proceedings of the IEEE/CVF International Conference on Computer Vision. 2023.
>
> [3] Beirami, Ahmad, et al. "Theoretical guarantees on the best-of-n alignment policy." arXiv preprint arXiv:2401.01879 (2024).

---

> > ### Author Response · Authors · 2025-03-30
> > **Response to Reviewer qWvG**
> >
> > > W1. Different denoising steps have varying impacts on the final output. The early stages primarily determine the overall structure of the image, while the later stages refine fine-grained details. CoDe could potentially improve its performance by incorporating adaptive control mechanisms that adjust sampling strategies based on the denoising stage.
> >
> > A: Thanks for your insightful  feedback. Following your suggestion, we perform an ablation experiment by varying $N, B$ across the first half $T_1 \in [500,250]$ and second half $T_2 \in (250,0]$ of the diffusion denoising process for the T2I compression guidance scenario in Appendix Section H, Table 10 of the revised manuscript. We experiment on a total of four cases by varying $N, B$ across the two halves of denoising. In the first two cases, $N=40$ throughout denoising and $B=1$ for $T_1$, $B=5$ for $T_2$ or $B=5$ for $T_1$, $B=1$ for $T_2$ (rows with $N=40, B_1=1, B_2=5$ and $N=40, B_1=5, B_2=1$ in Table 10). In the next two cases, $B=5$ throughout denoising and $N=40$ for $T_1$, $N=20$ for $T_2$ or $N=20$ for $T_1$, $N=40$ for $T_2$ (rows with $N_1=40, N_2=20, B=5$ and $N_1=20, N_2=40, B=5$ in Table 10). We observe that hyperparameters that enforce stronger reward-guidance in the second half of denoising ($T_2$) offer higher rewards as compared to the ones which enforce it in the first half ($T_1$). $N_1=40, N_2=20, B=5$ offers lower rewards, lower divergence (FID, CMMD) and on-par text alignment (T-CLIP) as compared to $N=[20,40], B=5$. Similar observations extend for ablating on $B$ where $N=40, B_1=1, B_2=5$ offers lower reward, lower divergence (FID, CMMD) and slightly lower text alignment (T-CLIP) as compared to $N=40, B_1=5, B_2=1$. Note that using a higher $N$ or lower $B$ throughout denoising to exert stronger inference-time guidance results in higher rewards and higher divergence as compared to their adaptive control counterparts. These observations for using stronger guidance in the second half of denoising intuitively align with the task of generating compressible images, where the fine-grained details and the texture of the image is to be controlled. But for other rewards that demand control over the structure and positioning of objects in generated images, a stronger reward-guidance signal in the first half of denoising might be more effective.

---

### Review · Reviewer_5ydM · 2025-02-26

**Summary Of Contributions:**

This paper explores the alignment between diffusion models and downstream tasks defined by rewards. The authors propose a blockwise sampling method applied during intermediate denoising steps, enabling alignment with downstream rewards. The key contributions of this work are:
1. An inference-time sampling technique based on controlled decoding, allowing reward evaluation at intermediate steps without requiring full denoising.
2. A blockwise approach that reduces the number of intermediate evaluations while maintaining generation quality.

**Audience:**

Yes

**Claims And Evidence:**

Yes

**Requested Changes:**

See  Strengths and Weaknesses

**Strengths And Weaknesses:**

Strengths:
1. The proposed method effectively aligns diffusion models with rewards. The authors validate their approach through experiments on both Gaussian Mixture Models and text-to-image diffusion models, demonstrating its effectiveness. The experimental results support their claims.
2. The proposed method is sound, and the paper is easy to follow.

Major Weaknesses:
1. There is a gap between the theory of controlled decoding and the final algorithm. Equation (6) suggests that the probability is proportional to $p\left(x_{t-1} \mid x_t\right) e^{\lambda V\left(x_{t-1} ; p\right)}$, where Algorithm 1 in the pseudo-code only considers  $V\left(x_{t-1} ; p\right)}$. From the final algorithm, it seems that the approach simply performs  x_0-prediction at each intermediate step and calculates the final reward based on  x_0, which is not explicitly reflected in the theoretical formulation.
2. The “best-of-N” strategy, which selects the sample with the highest reward, results in a lower I-Gram score compared to the baseline. This raises concerns about whether the reward is truly aligned with the evaluation metric, that is, if the evaluation metric can effectively show the alignment of the diffusion models.

Typos:
Several typos exist in the paper, such as “algnment” instead of “alignment” and “CoDeis” in the introduction section.

---

> ### Author Response · Authors · 2025-03-23
> **Response to Reviewer 5ydM**
>
> > W1. There is a gap between the theory of controlled decoding and the final algorithm. Equation (6) suggests that the probability is proportional to p(x_t-1 / x_t) exp(\lambda V(x_t-1; p)), where Algorithm 1 in the pseudo-code only considers V(x_{t-1} ; p)
>
> A1. Thanks for this feedback. We would like to clarify that the Equation (10) and not (6) from Section 2.2 suggests that the probability is proportional to $p(x_{t-1} / x_t) exp(\lambda V(x_t-1; p))$. We agree with you that there is a gap between the theory of controlled decoding in our equation and our blockwise decoding. However, we would like to mention that several recent works have shown that BoN is almost optimal in terms of reward (which is the value function in our case) vs divergence tradeoffs [3,4,5]. In particular, Theorem 1 from [4] shows that the samples obtained from BoN follow the same distribution as the optimal in from Eq. (10). This is the reason [8] reported the most favorable reward vs divergence tradeoffs using blockwise language model decoding as blockwise decoding is also almost optimal in terms of reward vs divergence given that it interpolates two (almost) optimal decoding schemes.
>
> Given Thm 2.1 and its proof in Appendix A, we aim to sample from the reward-tilted posterior $\pi_\lambda^*(x_{t-1}|x_{t}, c)$ in order to optimize the KL-regularized reward maximization objective Eq.9. In order to perform CoDe’s blockwise guidance, we:
>
> 1. sample from the prior $p(x_{t-1}|x_{t}, c)$ using the denoising diffusion process, across all $N$ streams,
>
> 2. and then compute the values of each of the $N$ samples using $V(x_{t-1} ; p)$
>
> By doing so, the probability of the selected sample $x_{t-1}$ with the highest value (in Alg.1 Line 8, Alg.2, Line 9) implicitly incorporates the prior distribution $p(x_{t-1}|x_{t}, c)$ as a Monte-Carlo estimation technique. Additionally, selecting the highest value sample:
>
> $x_{t-1} \gets argmax V(x_{t-1}^{(n)}; p, c)$
>
> is mathematically equivalent to sampling from the categorical distribution
>
> $x_{t-1} \overset {i.i.d.} \sim Categorical(\{\texttt{softmax}[V({x^{n}_{t-1}})/\tau]\}), \forall n \in [1,N],$
>
> where the temperature $\tau \to 0$. This technique for sampling from the posterior has also been used in related works such as [1].
>
> Following your remark and our clarifications, we have now included this explanation bridging the theory and practice of CoDe in Appendix C of the revised manuscript.
>
>
>
>
> > W2. From the final algorithm, it seems that the approach simply performs x_0-prediction at each intermediate step and calculates the final reward based on x_0, which is not explicitly reflected in the theoretical formulation.
>
> A2: Thanks for the remark. While the approximation of $V(x_{t}; p,c)$ using $r(\hat{x_0})$ is mentioned and discussed in Section 3, Eq.12, based on your feedback, we now explicitly mention this in Alg.1, Line 6 to estimate the values for each of the N samples $x_{t-1}^n$, and is also referred to in the explanation of the algorithm.

---

> ### Author Response · Authors · 2025-03-23
> **Response to Reviewer 5ydM**
>
> > W3.The “best-of-N” strategy, which selects the sample with the highest reward, results in a lower I-Gram score compared to the baseline. This raises concerns about whether the reward is truly aligned with the evaluation metric, that is, if the evaluation metric can effectively show the alignment of the diffusion models.
>
> A: Thanks for your insightful comment. The reason why we also use I-Gram to indicate reward-alignment (instead of only expected reward per scenario) in our evaluations is the train/test generalization since the expected reward has already been seen by the model throughout the guidance process. Given your spot-on remark we re-checked our analyses and have now uncovered and fixed a bug in the I-GRAM score reported for SDEdit($\eta$) baseline on the style guidance scenario in Table 5. This leads to a change in the I-GRAM scores of Table 2 as well, due to the normalization w.r.t. SDEdit and averaging across style, face and stroke guidance.
>
> Having said that, we do notice slight discrepancies in the behaviour of the I-GRAM scores and the CLIP-image similarity reward across a few generated samples due to the differences in what each of these metrics capture.
> The I-GRAM score has been shown to capture similarities in style and texture of two images [7] whereas the CLIP-image similarity score measures semantic similarity between any two images.
> We present a thorough evaluation across both these metrics to present a holistic analysis across all facets of performance.
> However, to confirm that I-GRAM can be used as an evaluation metric for reward guidance using CLIP similarity, we provide a qualitative analysis of the alignment in their behaviours across a few different generated images. This is illustrated in Appendix J, Fig.21 of the revised manuscript. Additionally, we computed the Pearson Correlation coefficient between the two metrics across a subset of all generated images in the style guidance scenario and observe a correlation coefficient of $0.87$ between Reward and I-GRAM, indicating a positive correlation / direct proportionality between the two.
>
>
> References:
>
> [1] Li, Xiner, et al. "Derivative-free guidance in continuous and discrete diffusion models with soft value-based decoding." arXiv preprint arXiv:2408.08252 (2024).
>
> [2] Uehara, Masatoshi, et al. "Inference-Time Alignment in Diffusion Models with Reward-Guided Generation: Tutorial and Review." arXiv preprint arXiv:2501.09685 (2025).
>
> [3] Beirami, Ahmad, et al. "Theoretical guarantees on the best-of-n alignment policy." arXiv preprint arXiv:2401.01879 (2024).
>
> [4] Yang, Joy Qiping, et al. "Asymptotics of language model alignment." 2024 IEEE International Symposium on Information Theory (ISIT). IEEE, 2024.
>
> [5] Gui, Lin, Cristina Gârbacea, and Victor Veitch. "Bonbon alignment for large language models and the sweetness of best-of-n sampling." NeurIPS (2024).
>
> [7] Gatys, Leon A., Alexander S. Ecker, and Matthias Bethge. "A neural algorithm of artistic style." arXiv preprint arXiv:1508.06576 (2015).
>
> [8] Mudgal, Sidharth, et al. "Controlled decoding from language models." ICML (2024).

---

### Review · Reviewer_2j8L · 2025-02-28

**Summary Of Contributions:**

The blockwise approach bridges the gap between BoN and stepwise methods, offering a flexible balance between computational cost and reward alignment. This paper provides a concise algorithmic implementation along with theoretical proofs to demonstrate the effectiveness of blockwise sampling methods. It evaluates performance on tasks such as style transfer, image editing, and GMM model fitting using metrics like I-Gram, FID, T-CLIP, and runtime.

**Audience:**

Yes

**Broader Impact Concerns:**

No more concerns.

**Claims And Evidence:**

Yes

**Requested Changes:**

In the experiments involving quantitative metrics, the hyperparameters are not clearly specified. However, there are many aspects that can be tuned, and these additional searches could potentially improve performance. The paper should clearly indicate these, and ensure that the runtime in Table 2 remains consistent for a fair comparison.

**Strengths And Weaknesses:**

*Strengths*:
1. **Novelty**: The blockwise approach bridges the gap between BoN and stepwise methods, offering a flexible balance between computational cost and reward alignment.
2. **Theoretical rigor**: The connection to KL-regularized objectives provides a solid foundation, and the analysis of approximation error (via Tweedie’s formula) is well-motivated.
3. **Comprehensive evaluation**: Experiments span synthetic (GMM) and real-world (text-to-image, text-image-to-image) tasks, with both differentiable and non-differentiable rewards, showcasing broad applicability.
4. **Practical insights**: The noise conditioning mechanism (CoDe(η)) and ablation studies on block size (B) and η provide actionable guidelines for practitioners.

*Weaknesses*:
1. The method proposed in this paper is relatively straightforward; the truncation at each step does not seem very innovative, and the overall contribution is not particularly substantial.
2. As a work primarily focused on providing tradeoffs, it should more comprehensively present the performance–efficiency tradeoff curve space to clearly illustrate the actual contribution. However, the paper only reports efficiency tests under fixed settings, and it does not disclose the search cost used in performance comparisons, which leaves readers unclear about the actual contribution.
3. The experimental tasks selected in the paper are relatively simple.

---

> ### Author Response · Authors · 2025-03-23
> **Response to Reviewer 2j8L**
>
> > W1. The method proposed in this paper is relatively straightforward; the truncation at each step does not seem very innovative, and the overall contribution is not particularly substantial.
>
> A1: We agree that our proposed method is simple. That being said, the key aim of CoDe is to present a simple yet competitive inference-time guidance method as compared to its more computationally complex counterparts, across multiple reward models. Having said that, we would like to also mention that to the best of our knowledge that our work is the first to use blockwise decoding in denoising diffusion models even though such schemes had been used for decoding of language models.
>
>
> > W2. As a work primarily focused on providing tradeoffs, it should more comprehensively present the performance–efficiency tradeoff curve space to clearly illustrate the actual contribution. However, the paper only reports efficiency tests under fixed settings, and it does not disclose the search cost used in performance comparisons, which leaves readers unclear about the actual contribution. **And** Clearly specifying hyperparameters.
>
> A: Thanks for this feedback. Following your suggestion, we now also present the reward vs compute and FID vs compute tradeoff curves in Section 6.5 - Performance vs Efficiency Tradeoffs in Fig.10. We observe that CoDe($\eta$) and SVDD-PM($\eta$) offer better reward vs compute compared to all other baselines. Specifically, while SVDD-PM($\eta$) achieves higher rewards for the same compute as compared to CoDe($\eta$), it also deviates significantly more from the base distribution as compared to CoDe($\eta$) – where divergence captures preserving core capabilities not captured by reward – resulting in inferior reward vs divergence tradeoffs that are discussed in the previous sections. Thus, in terms of a tradeoff between performance (captured through reward and divergence) vs efficiency, CoDe($\eta$) still offers a better tradeoff as compared to SVDD-PM($\eta$) enabling performance points that are not even achievable by SVDDPM. On the other hand, UG offers high rewards but at the cost of either significantly higher compute or FID. Also, following your remark on clearly specifying the hyperparameters being used across different scenarios and settings, we have now mentioned these in Appendix E of the revised manuscript. This is in addition to the details of the sweeping range of parameters used for the tradeoff and ablation curves, mentioned in Sections 6.1, 6.3.
>
> Additionally, we would like to highlight some of the other existing comparisons for a full picture:
>
> i. The breakdown of total computational costs in terms of inference steps and reward model queries have been included in Table 5 of the manuscript. These have been expressed in terms of the corresponding method’s hyperparameters (for eg: N,B,T,\eta for CoDe(\eta), N,T for BoN). This enables the reader to calculate computational complexity (runtime) for any point across all trade-off curves.
>
> ii.The runtime for each method averaged across all the differentiable rewards (in seconds/image generated).
>
> iii. A detailed breakdown of the runtime for each method across all the differentiable rewards (in seconds/image generated) in Appendix E, Table 9.
>
> iv. The reward/performance - divergence curves for - a. non-differentiable reward (compressibility guidance) is included in Fig.5; b. Differentiable rewards are included in Figs.13,14,18.
>
>
> > W3. The experimental tasks selected in the paper are relatively simple.
>
> A: Thanks for constructive feedback. Following your suggestion we have now experimented with Aesthetic score based reward guidance [1] and compared against sampling based (BoN) and gradient based baselines (UG, MPGD, Freedom). The results for this have now been added in the Section 6.2, Table 3 and the reward vs divergence, reward vs text-alignment trade-off curves have been added in Fig.18 of the revised manuscript.
>
> The aesthetic reward has been trained on a preference dataset of aesthetically stylized and simple/natural images and thus could have multiple optimal solutions(generated images) for a given prompt.
>
> This offers a different perspective from our previous differentiable rewards (style, face, stroke transfer) where the CLIP image similarity reward is maximized only when the generated image completely mimics the target image (CLIP[generated image = target image, target image]=1).
>
> We observe that CoDe offers better rewards as compared to MPGD and BoN while being second to best as compared to Freedom and UG . However, CoDe offers better text alignment (T-CLIP) and a significantly lower divergence from the base distribution (FID, CMMD) as compared to all its gradient based counterparts, thus offering a competitive reward vs divergence tradeoff. This can also be observed in Fig.18, where CoDe offers almost the same reward as Freedom and UG, but at a lower divergence or higher T-CLIP, which indicates better performance overall.

---

> > ### Author Response · Authors · 2025-03-23
> > **Response to Reviewer 2j8L**
> >
> > References:
> >
> > [1] Schuhmann, Christoph, et al. "Laion-5b: An open large-scale dataset for training next generation image-text models." Advances in neural information processing systems 35 (2022): 25278-25294.

---

### Author Response · Authors · 2025-03-23
**Summary of Responses**

Dear reviewers and the AC,

We appreciate your constructive feedback which helped us further improve the quality and clarity of the paper. We are pleased by the positive feedback from reviewer 2j8L for finding our proposed method "theoretically rigorous", "novel" and our experimental analyses practically insightful, reviewer 5yDM for finding our paper "well-written and easy to follow" and reviewer qWvG for finding CoDe “versatile for real world tasks”.

We would like to summarize the revisions we have made in the rebuttal period to address the reviewers’ remarks as follows:

1. In response to reviewer **2j8L**, we expand on the performance vs efficiency tradeoffs: we now present 2 new tradeoff curves:
Reward vs Compute and FID vs Compute comparing multiple sampling based (CoDe, BoN, SVDD-PM) and gradient based (UG, DPS) guidance baselines, across their respective tuning hyperparameters. These new tradeoff curves are illustrated in Fig.10, Section 6.5 - Performance vs Efficiency Tradeoffs of the revised manuscript. We observe that CoDe($\eta$) and SVDD-PM($\eta$) offer the best reward vs compute and FID vs compute tradeoffs as compared to all other baselines. Specifically, while SVDD-PM($\eta$) achieves higher rewards for the same compute as compared to CoDe($\eta$), it also deviates significantly more from the base distribution as compared to CoDe($\eta$). Thus, in terms of a full 3-dimensional tradeoff between reward performance, divergence (which captures preserving other core capabilities not captured by the reward) and efficiency, CoDe($\eta$) offers a better tradeoff as compared to SVDD-PM($\eta$). On the other hand, UG offers high rewards but at the cost of either significantly higher compute or FID.


2. In response to reviewers **2j8L** and **qWvG** remarks on the simplicity of the chosen guidance tasks and including comparisons against more advanced gradient based methods, respectively, we now present guidance results on a new T2I guidance task - Aesthetic score based reward guidance. Here, we also compare CoDe’s performance against two recent advanced gradient based baselines MPGD and Freedom. The results for this have now been added in the Appendix Section 6.2, Table 3 and the reward vs divergence, reward vs text-alignment trade-off curves have been added in Fig.18 of the revised manuscript. We observe that CoDe offers better rewards as compared to MPGD and BoN while being second to best as compared to Freedom and UG . However, CoDe offers better text alignment (T-CLIP) and a significantly lower divergence from the base distribution (FID, CMMD) as compared to all its gradient based counterparts, thus offering a better overall reward vs divergence tradeoff. This can also be observed in Fig.18, where CoDe offers almost the same reward as Freedom and UG, but at a lower divergence or higher T-CLIP.


3. Following reviewer **5yDM**’s remark on a discrepancy between our guidance-reward metric (CLIP image similarity) and evaluation metric (I-GRAM ), we uncovered a bug in the calculation of the I-GRAM score reported for SDEdit($\eta$) baseline on the style guidance scenario in Table 6. Unfortunately, this error propagated to the I-GRAM scores of Table 2 as well, due to the normalization w.r.t. SDEdit and averaging across style, face and stroke guidance. We have now fixed this bug and corrected all the erroneous numbers. Additionally, to further analyse the discrepancies between I-GRAM and our reward, we now include a few qualitative demonstrations of cases where the two metrics align vs are slightly misaligned in Appendix J, Fig.21. We hope this helps the readers intuitively understand the subtle differences between the two metrics. To quantitatively corroborate that I-GRAM effectively can be used as an evaluation metric for style, face and stroke guidance scenarios, we compute the Pearson correlation coefficient between the two and observe a value of $0.87$, which indicates  a reasonable positive correlation.


**Remark**: Note that the additional experimental results and explanations have been added to the revised manuscript in $\textcolor{blue}{blue}$.
We do hope our additional experiments and responses have addressed all the reviewers' concerns and questions.

---

> ### Author Response · Authors · 2025-03-30
> **Summary of Responses**
>
> 4. In response to reviewer **qWvG**’s remark on using an adaptive control strategy on $N, B$ for CoDe’s blockwise guidance, we perform an ablation experiment by varying $N, B$ across the first half $T_1 \in [500,250]$ and second half $T_2 \in (250,0]$ of the diffusion denoising process for the T2I compression guidance scenario, in appendix section H, Table 10 of the revised manuscript. We observe that hyperparameters that enforce stronger reward-guidance in the second half of denoising ($T_2$) offer higher rewards as compared to the ones which enforce it in the first half ($T_1$). $N_1=40, N_2=20, B=5$ offers lower rewards, lower divergence (FID, CMMD) and on-par text alignment (T-CLIP) as compared to $N=[20,40], B=5$. Similar observations extend for ablating on $B$ where $N=40, B_1=1, B_2=5$ offers lower reward, lower divergence (FID, CMMD) and slightly lower text alignment (T-CLIP) as compared to $N=40, B_1=5, B_2=1$. These observations for using stronger guidance in the second half of denoising intuitively align with the task of generating compressible images, where the fine-grained details and the texture of the image is to be controlled. But for other rewards that demand control over the structure and positioning of objects in generated images, a stronger reward-guidance signal in the first half of denoising might be more effective.

---

### Decision · Action_Editor_2QeA · 2025-04-14

**Recommendation:** Accept with minor revision

**Comment:**

The paper receives mixed reviews. Two leaning accepts and one leaning rejects. The main critiques include lack of novelty, need of more comprehensive evaluations and a potential issue related to the I-Gram score evaluation. The authors provided extensive responses, also fixed a bug in their evaluation which I believe addressed some of the major concerns. I'm also leaning to down weigh the importance of novelty as its flip side is simplicity which is a strong suit on its own. Overall, I think this work is worth including in TMLR with a minor revision by incorporating the reviewer feedbacks.

**Audience:**

Alignment is an important topic in generative models, and diffusion models in specific. Therefore, this work is relevant to the broad generative modeling community.

**Claims And Evidence:**

This work provides a simple blockwise control for aligning pretrained diffusion models to various reward functions. The claim is that the proposed technique achieves favorable tradeoffs between the quality and cost of alignment, and it is evaluated in several representative reward settings.